# ONLINE MULTI-OBJECTIVE CONVEX OPTIMIZATION: A UNIFIED FRAMEWORK AND JOINT GRADIENT DESCENT

## ABSTRACT

Online Convex Optimization (OCO) usually addresses the learning task with a single objective; however, in real-world applications, multiple conflicting objectives often need to be optimized simultaneously. In this paper, we present an Online Multi-objective Convex Optimization (OMCO) framework with a novel multi-objective regret. We prove that, when the number of objectives in OMCO decreases to one, the regret is equal to the regret in OCO, thus unifying the OCO and OMCO frameworks. To facilitate the analysis of the proposed novel regret, we derive its equivalent form using the strong duality theory of convex optimization. Moreover, we propose an Online Joint Gradient Descent algorithm and prove that it achieves a sublinear multi-objective regret according to the equivalent regret form. Experimental results on several real-world datasets validate the effectiveness of our proposed algorithm.

## 1 INTRODUCTION

In recent years, with the rapid advancement of the Internet industry and Internet of Things technologies, online learning has become an essential paradigm for efficiently processing streaming data (Hoi et al., 2021). In online learning, the paramount framework is Online Convex Optimization (OCO) (Hazan, 2022), which has been widely studied. OCO can be regarded as a repeated game between a learner and an adversary: at each $t$-th round, the learner makes a decision $\boldsymbol{x}_t$ from a convex set $X \in \mathbb{R}^n$ for future unforeseeable data based on previous data, and then the adversary reveals a convex loss $f_t(\boldsymbol{x}_t)$, which is unknown in advance and may change with time. The goal of the learner is to minimize $regret$, defined as the difference between the cumulative loss of the learner and that of a comparator sequence in hindsight over the time horizon $T$, i.e.,

$$regret_T = \sum_{t=1}^{T} f_t(\boldsymbol{x}_t) - \min_{\boldsymbol{x} \in X} \sum_{t=1}^{T} f_t(\boldsymbol{x}). \tag{1}$$

The regret in the OCO framework implies that: if an algorithm incurs regret that grows sub-linearly with time, its performance loss tends to zero when averaged over an infinite time horizon $T$.

Under the OCO framework, many OCO algorithms have been developed (Zinkevich, 2003; Hazan et al., 2007; Zhang et al., 2019; Wang et al., 2020), which are designed based on a scalar loss $f_t$ revealed by an adversary, i.e., single-objective cases. However, we often need to consider multiple objectives simultaneously in real-world applications. For example, in natural language processing (Li et al., 2021; Chen et al., 2021), multiple objectives such as sequence tagging, classification, and text generation, are required to be optimized simultaneously. In autonomous driving (Chen et al., 2018; Ravindran et al., 2020), the vehicle needs to solve object detection and distance prediction at the same time. In these cases, when the revealed losses are vectors and convex, these problems are called Online Multi-objective Convex Optimization (OMCO) problems.

To address OMCO problems, most existing work transforms the multi-objective optimization problem into a single-objective form by using scalarization techniques for further analysis. The most common scalarization method is to combine multiple objectives linearly (Cesa-Bianchi et al., 2022; Achddou et al., 2024; Busa-Fekete et al., 2017; Liu et al., 2024). Besides, methodologies based on

$\epsilon$-constraint (Mahdavi et al., 2013) and reference point techniques have been investigated (Lyu et al., 2019; 2024). However, for the above algorithms, the single-objective functions obtained by different scalarization techniques and the counterpart regret forms are different. These regrets have different meanings for an OMCO problem and the obtained regret bounds cannot be compared. In addition, these regret cannot adequately reflect the trade-off relationship and the interaction among multiple objectives.

In fact, unlike a single-objective optimization problem, a multi-objective optimization problem has no unique optimal solution, but a set of efficient solutions that do not dominate each other (Ehrgott, 2005; Caramia et al., 2020). The image of these solutions in the objective space is called the non-dominated set. There are a few works that have considered OMCO problems from a multi-objective perspective (Blackwell, 1956; Mannor et al., 2014; Tiedemann et al., 2015; Turgay et al., 2018). Recently, Jiang et al. (2023) proposed a OMCO framework and a multi-objective regret called Sequence-wise Pareto Suboptimality Gap (S-PSG). S-PSG measures the discrepancy from the cumulative loss of the online decisions to the non-dominated set. However, S-PSG restricted to be non-negative. It needs to be noted that the regret in the OCO framework may be less than zero if a decision maker happens to make an optimal decision at each round. Considering the uniformity of the online convex optimization framework for any number of objectives, the non-negativity of S-PSG is inconsistent with the regret (which can be negative) defined in the OCO framework. This raises a natural question: *Can we develop a generalized OMCO framework with a designed multi-objective regret such that it has the following properties?*

- The designed multi-objective regret can reflect the relationship between the accumulated vector loss and the non-dominated set of an OMCO problem in hindsight.

- The OMCO framework is a generalized version of OCO, i.e., when the number of objectives decreases to one, the multi-objective regret degenerates into the regret of OCO.

In this paper, we give a positive answer to the above question. We define a novel multi-objective regret, which is the difference between the cumulative loss vector and the (weakly) non-dominated set along the direction of $e$ vector, where $e$ is a vector in which each entry is 1. This difference can be obtained by solving a Translative Scalarization (TS) problem (Pascoletti & Serafini, 1984). We prove that this multi-objective regret degenerates to the regret in the OCO framework when the number of objectives is 1, thus unifying the online convex optimization framework for any number of objectives. To further support the analysis of OMCO algorithms, we derive an equivalent form of the regret in the OMCO framework using the strong duality theory of convex optimization. Furthermore, we consider minimizing the worst-case objective value and propose an Online Joint Gradient Descent (OJGD) algorithm. In the algorithm, we introduce weight variables for instantaneous loss functions to construct a joint gradient descent direction, and the weights are updated by an ascent step. Compared to the methods based on minimizing the (regularized) norm of the joint gradient (Jiang et al., 2023; Désidéri, 2012; Sener & Koltun, 2018), our proposed algorithm is easier to implement and updates the weights in linear time. We prove that the OJGD algorithm guarantees a sublinear multi-objective regret.

To validate the effectiveness of our proposed algorithm, we provide experimental evaluations on two classical machine learning convex problem models (i.e., linear regression and logistic regression problem models). Furthermore, to verify OJGD's generalizability, we conduct experiments on multi-task learning (MTL) networks with non-convex objective functions, which essentially represent non-convex multi-objective optimization problems (Sener & Koltun, 2018; Liu et al., 2019). We use numerical results to show the superior performance of our proposed algorithm on several real-world datasets. Our main contributions are as follows.

- **Unified optimization framework.** We establish an OMCO framework with a novel multi-objective regret and prove that the proposed novel regret degenerates to the regret in the OCO framework when the number of objectives is equal to one, thereby unifying the online convex optimization framework for any number of objectives.

- **OMCO algorithm with robustness.** Considering minimizing the worst-case objective value, we propose an OMCO algorithm called Online Joint Gradient Descent, which is easy to implement and is proved to guarantee a sublinear multi-objective regret.

- **Experiments on practical applications.** We provide empirical evaluations of our proposed algorithm on several real-world datasets. The results show that our algorithm consistently outperforms other OMCO algorithms.

## 2 RELATED WORKS

**Online multi-objective convex optimization.** Among those methods that transform the OMCO problem into single-objective forms, the most straightforward method is to sum multiple objectives (Cesa-Bianchi et al., 2022; Achddou et al., 2024). Besides, weighted sum methods have been studied to convexly combine multiple objectives into a single objective (Busa-Fekete et al., 2017; Liu et al., 2024). Mahdavi et al. (2013) used the $\epsilon$-constraint method (Ehrgott, 2005) to choose a primary objective as the objective function with other objectives bounded by thresholds. Based on an expected point in the objective space, Lyu et al. (2019; 2024) established a quadratic programming problem to minimize the distance between the average cumulative loss and the expected point. Furthermore, a few works have studied OMCO problems from a multi-objective perspective. Blackwell (1956) and Mannor et al. (2014) considered that the learner's goal is to make the average vectorial loss function converge to a closed target set known in advance. However, for OMCO problems, the target set is a set of the (weakly) non-dominated points, which is unknown in advance. Tiedemann et al. (2015) introduced a framework for the competitive analysis of OMCO algorithms. Turgay et al. (2018) proposed a metric called Pareto Suboptimality Gap (PSG) for multi-objective contextual bandit problems, where the PSG is restricted to be non-negative. Jiang et al. (2023) generalized PSG to OMCO problems and defined the S-PSG multi-objective regret of OMCO. The S-PSG is also restricted to a non-negative domain, which exposes a fundamental inconsistency with the regret in the OCO framework, allowing the sign to be unrestricted. In this paper, we propose a unified online convex optimization framework for any number of objectives.

**Multi-objective optimization.** There are two classes of multi-objective optimization (MO) methods that are relevant to our work, i.e., translative scalarization methods (Pascoletti & Serafini, 1984; Benson & Sayin, 1997) and gradient-based methods (Fliege & Svaiter, 2000; Povalej, 2014). In a translative scalarization method, a single-objective optimization problem is constructed based on a given reference point and a projection direction in the objective space. By solving the problem, the projection of this reference point on the (weakly) non-dominated set is obtained along this projection direction. Many MO methods have been designed based on this technique (Das & Dennis, 1998; Ismail-Yahaya & Messac, 2002; Shao & Ehrgott, 2016). In this paper, we design a novel multi-objective regret based on this technique. Gradient-based methods were designed to solve gradient conflict problems in MO (Désidéri, 2012; Sener & Koltun, 2018; Yu et al., 2020; Liu et al., 2021). The min-norm method (Désidéri, 2012; Sener & Koltun, 2018), which obtains the gradient weights by minimizing the norm of the joint gradient, is a foundational algorithm in this regard. However, most of the above MO methods are designed for offline settings and non-adversarial MO problems. In recent years, Jiang et al. (2023) extended such a min-norm method to OMCO problems and proposed the Doubly Regularized Online Mirror Multiple Descent (DR-OMMD) algorithm. The DR-OMMD algorithm obtains the weights of the joint gradient by solving a quadratic optimization problem with a regularization term, which incurs substantial computational overhead in practice. In this paper, our proposed OJGD uses an ascent step to update the weights of the joint gradient, which can be implemented in linear time.

## 3 MULTI-OBJECTIVE CONVEX OPTIMIZATION

Throughout the paper vectors are column vectors written in boldface. For a vector $\boldsymbol{x} \in \mathbb{R}^p$, we write $\boldsymbol{x}^i$ to denote the $i$-th component of the vector $\boldsymbol{x}$. We denote by $\boldsymbol{x}^T\boldsymbol{y}$ the inner product of two vectors $\boldsymbol{x}$ and $\boldsymbol{y}$. We use $\|\boldsymbol{x}\|$ to denote the standard Euclidean norm, i.e., $\|\boldsymbol{x}\| = \sqrt{\boldsymbol{x}^T\boldsymbol{x}}$. For $p \in \mathbb{N}$, we denote by $[p]$ as the set $\{1, 2, \cdots, p\}$. We denote by $\boldsymbol{e} \in \mathbb{R}^p$ as a vector in which each entry is 1. Furthermore, $\boldsymbol{e}_i$ denotes a vector whose $i$-th entry is 1 and others are 0. We use $\Pi_X(\boldsymbol{y})$ to denote the Euclidean projection of a vector $\boldsymbol{y}$ onto $X$, i.e., $\Pi_X(\boldsymbol{y}) = \operatorname{argmin}_{\boldsymbol{x} \in X}\|\boldsymbol{x} - \boldsymbol{y}\|$. Let $\Lambda := \{\boldsymbol{\lambda} \in \mathbb{R}^p \mid \forall i \in [p], \boldsymbol{\lambda}^i \geqslant 0, \boldsymbol{e}^T\boldsymbol{\lambda} = 1\}$ and $\mathbb{R}^p_{\geqslant} := \{\boldsymbol{x} \in \mathbb{R}^p \mid \forall i \in [p], \boldsymbol{x}^i \geqslant 0\}$. For a set $\mathcal{S} \subseteq \mathbb{R}^p$, we denote the interior of $\mathcal{S}$ by $\text{int}\mathcal{S}$.

**Definition 1.** *For a set $\mathcal{S} \subseteq \mathbb{R}^p$, we define its non-dominated set $\mathcal{S}_N$ and its weakly non-dominated set $\mathcal{S}_{WN}$ as follows.*

$$\mathcal{S}_N := \left\{ \boldsymbol{y} \in \mathcal{S} : (\{\boldsymbol{y}\} - \mathbb{R}^p_{\geqslant}) \backslash \{0\} \cap \mathcal{S} = \emptyset \right\}, \quad \mathcal{S}_{WN} := \left\{ \boldsymbol{y} \in \mathcal{S} : (\{\boldsymbol{y}\} - int\mathbb{R}^p_{\geqslant}) \cap \mathcal{S} = \emptyset \right\}.$$

**Definition 2.** *For two sets $\mathcal{A} \subseteq \mathbb{R}^p$ and $\mathcal{B} \subseteq \mathbb{R}^p$, their Minkowski sum is defined by $\mathcal{A} \oplus \mathcal{B} = \{\boldsymbol{x} + \boldsymbol{y} | \boldsymbol{x} \in \mathcal{A}, \boldsymbol{y} \in \mathcal{B}\}$. For a set $\mathcal{S} \subseteq \mathbb{R}^p$, the $\mathbb{R}^p_{\geqslant}$ cone expansion of $\mathcal{S}$ is defined by $\mathcal{S}' := \mathcal{S} \oplus \mathbb{R}^p_{\geqslant}$.*

According to Definitions 1 and 2, we have $S_N = S'_N$ and $S'_N \subseteq S'_{WN}$. Then, we consider a multi-objective convex optimization (MCO) problem:

$$\min_{\boldsymbol{x} \in X} \boldsymbol{F}(\boldsymbol{x}) = (F^1(\boldsymbol{x}), F^2(\boldsymbol{x}), \cdots, F^p(\boldsymbol{x}))^T, \tag{2}$$

where $X$ is a non-empty convex compact set in decision (or variable) space $\mathbb{R}^n$. $\boldsymbol{F} : \mathbb{R}^n \to \mathbb{R}^p$ is a vector loss composed of $p$ scalar losses, i.e., $\boldsymbol{F} = (F^1, \cdots, F^p)^T$, where $\forall i \in [p], F^i$ is convex and differentiable. The image of the feasible set $X$ under the vector loss $\boldsymbol{F}$ is denoted as $Y := \boldsymbol{F}(X) \subset \mathbb{R}^p$, it is called the feasible set in objective space.

The set of optimal solutions for an MCO problem (2) is actually a set of efficient solutions. Specifically, if $\bar{\boldsymbol{x}} \in X$ is an efficient solution of an MCO problem (2), then for the image $\boldsymbol{F}(\bar{\boldsymbol{x}})$ of $\bar{\boldsymbol{x}}$ in the objective space, there are no other feasible points that can simultaneously improve all objective values of $\boldsymbol{F}(\bar{\boldsymbol{x}})$. The mathematical definition of an (weakly) effective solution is as follows.

**Definition 3** (Ehrgott, 2005). *For an MCO problem (2) and a feasible set $X$ in the decision space, a solution $\boldsymbol{x} \in X$ is called a (weakly) efficient solution if $(\boldsymbol{F}(\boldsymbol{x}) \in Y_{WN})$ $\boldsymbol{F}(\boldsymbol{x}) \in Y_N$. Correspondingly, point $\boldsymbol{y} = \boldsymbol{F}(\boldsymbol{x})$ is called a (weakly) non-dominated point in the objective space, where the set of non-dominated points is called the Pareto front.*

The **translative scalarization** (Pascoletti & Serafini, 1984) problem of an MCO problem (2) is

$$\max \delta, \ s.t. \ \boldsymbol{q} - \delta\boldsymbol{e} \geqslant \boldsymbol{F}(\boldsymbol{x}), \ s.t. \ \boldsymbol{x} \in X, \ \delta \in \mathbb{R}, \tag{3}$$

where $\boldsymbol{q} \in \mathbb{R}^p$ is a given reference point.

**Lemma 1** (Pascoletti & Serafini, 1984). *Given a reference point $\boldsymbol{q} \in \mathbb{R}^p$, suppose $\bar{\boldsymbol{x}} \in X$ is an optimal solution of the translative scalarization problem (3), then $\bar{\boldsymbol{x}}$ is a weakly efficient solution of an MCO problem (2) and its image in the objective space is a weakly non-dominated point.*

In fact, the optimal value $\delta^*$ of problem (3) is the difference between $\boldsymbol{q}$ and $Y'$ along the direction $\boldsymbol{e}$ and this difference could be either non-negative or negative depending on whether $\boldsymbol{q} \in Y'$ or not. In the next section, we will use this translative scalarization to establish an OMCO framework.

## 4 ONLINE MULTI-OBJECTIVE CONVEX OPTIMIZATION

We now gradually formulate the proposed OMCO framework. Similar to the OCO framework for single-objective cases, OMCO can be described as a game process between a learner and an adversary: at each round $t \in [T]$, the learner first makes a decision $\boldsymbol{x}_t \in X$ for arriving data. Then the adversary reveals a vector $\boldsymbol{F}_t : X \to \mathbb{R}^p$ of loss functions, and the learner suffers the vector-valued loss $\boldsymbol{F}_t(\boldsymbol{x}_t)$. The learner's goal is to minimize a certain kind of regret.

For an OMCO problem, we are interested in obtaining a (weak) effective solution to the following problem.

$$\min \sum_{t=1}^{T} \boldsymbol{F}_t(\boldsymbol{x}) = (\sum_{t=1}^{T} \boldsymbol{F}_t^1(\boldsymbol{x}), \cdots, \sum_{t=1}^{T} \boldsymbol{F}_t^p(\boldsymbol{x})), \ s.t. \ \boldsymbol{x} \in X. \tag{4}$$

In the literature, problem (4) is often transformed into a single-objective online optimization problem with weights $\boldsymbol{\lambda} \in \Lambda$ by the weighted sum method, i.e., $(P_\lambda) : \min_{\boldsymbol{x} \in X} \boldsymbol{\lambda}^T \sum_{t=1}^{T} \boldsymbol{F}_t(\boldsymbol{x})$. Correspondingly, the weighted sum regret for fixed $\boldsymbol{\lambda}$ is $regret_T(\boldsymbol{\lambda}) = \boldsymbol{\lambda}^T \sum_{t=1}^{T} \boldsymbol{F}_t(\boldsymbol{x}_t) - \min_{\boldsymbol{x} \in X} \boldsymbol{\lambda}^T \sum_{t=1}^{T} \boldsymbol{F}_t(\boldsymbol{x})$. However, in the weighted sum method, not only a priori knowledge is required to determine the weights, but the trade-off relationship and the interaction among multiple objectives cannot be adequately reflected. To extend the regret of the OCO framework to multi-objective cases, based on the translative scalarization method (3), we define a novel multi-objective regret as follows.

**Definition 4.** *The multi-objective regret in the OMCO framework is*

$$R_T = \max_{\boldsymbol{x} \in X, \delta \in \mathbb{R}} \{\delta : \sum_{t=1}^{T} \boldsymbol{F}_t(\boldsymbol{x}_t) - \delta \boldsymbol{e} \geqslant \sum_{t=1}^{T} \boldsymbol{F}_t(\boldsymbol{x})\}. \tag{5}$$

Let $Y'$ be the cone expansion of the feasible set $Y$ in the objective space for an MCO problem (4) and $(\delta^*, \boldsymbol{x}^*)$ be an optimal solution of problem (5). It should be noted that $\delta^*$ actually measures the difference between the cumulative loss and $Y'_{WN}$ in the $\boldsymbol{e}$-direction. Here, $\delta^*$ could be non-negative or negative depending on whether $\sum_{t=1}^{T} \boldsymbol{F}_t(\boldsymbol{x}_t) \in Y'$ or not. Also, $\boldsymbol{x}^*$ is a (weakly) efficient solution and $\sum_{t=1}^{T} \boldsymbol{F}_t(\boldsymbol{x}^*)$ is a (weakly) non-dominated point of the problem (4). Moreover, $R_T$ and $regret_T$ have the following relationship.

**Theorem 1.** *When the number of objectives is degenerated to one, $R_T$ is equal to the regret in the OCO framework, i.e., $R_T = regret_T$, when $p = 1$.*

*Proof.* When $p = 1$, we have $R_T = \max_{\boldsymbol{x} \in X, \delta \in \mathbb{R}} \{\delta : \sum_{t=1}^{T} f_t(\boldsymbol{x}_t) - \delta \geqslant \sum_{t=1}^{T} f_t(\boldsymbol{x})\} = \max_{\boldsymbol{x} \in X}(\sum_{t=1}^{T} f_t(\boldsymbol{x}_t) - \sum_{t=1}^{T} f_t(\boldsymbol{x})) = \sum_{t=1}^{T} f_t(\boldsymbol{x}_t) - \min_{\boldsymbol{x} \in X} \sum_{t=1}^{T} f_t(\boldsymbol{x}) = regret_T$. Therefore, Theorem 1 holds. $\square$

**Remark 1.** *According to Theorem 1, the regret in single-objective cases is actually a degenerate version of the regret in multi-objective cases. Therefore, we can use '$R_T$' to denote the regret of the online convex optimization framework with any number of objectives.*

For the form of regret (5) in the OMCO framework, it is difficult to directly guide the design of OMCO algorithms and analyze their properties. Therefore, we next derive its equivalent form as shown in Theorem 2.

**Theorem 2.** *Assume that there exists $\delta_0 \in \mathbb{R}$ and $\boldsymbol{x}_0 \in intX$ such that $\sum_{t=1}^{T} \boldsymbol{F}_t(\boldsymbol{x}_t) - \delta_0 \boldsymbol{e} > \sum_{t=1}^{T} \boldsymbol{F}_t(\boldsymbol{x}_0)$ holds, then an equivalent form of the multi-objective regret (5) is*

$$R_T = \min_{\boldsymbol{\lambda} \in \Lambda} \max_{\boldsymbol{x} \in X} \boldsymbol{\lambda}^T (\sum_{t=1}^{T} \boldsymbol{F}_t(\boldsymbol{x}_t) - \sum_{t=1}^{T} \boldsymbol{F}_t(\boldsymbol{x})). \tag{6}$$

*Proof.* Problem (5) is a convex optimization problem and its Lagrangian dual problem is

$$\min_{\boldsymbol{\lambda} \in \mathbb{R}_{\geqslant}} \max_{\boldsymbol{x} \in X, \delta \in \mathbb{R}} \delta + \boldsymbol{\lambda}^T (\sum_{t=1}^{T} \boldsymbol{F}_t(\boldsymbol{x}_t) - \delta \boldsymbol{e} - \sum_{t=1}^{T} \boldsymbol{F}_t(\boldsymbol{x})), \tag{7}$$

where $\boldsymbol{\lambda}$ is the dual variable. According to the assumption, it is known that Slater's condition is satisfied. Thus, the problem pair (5) and (7) has strong duality. Consequently, the optimal values of (5) and (7) are equal. Furthermore, by the KKT conditions, the derivative of objective function of problem (7) with respect to $\delta$ is zero, we have $\boldsymbol{\lambda} \in \Lambda$. Then, problem (7) is equivalent to the right-hand side of (6). Therefore, Theorem 2 holds. $\square$

Based on the regret form in Theorem 2, we can easily find that this regret is strongly related to the weighted sum form of OMCO problem, which is shown in Proposition 1 and 2.

**Proposition 1.** *Suppose that $(\boldsymbol{x}^*, \boldsymbol{\lambda}^*)$ is an optimal solution of the above problem (6), if an OMCO problem is transformed into an OCO problem $(P_\lambda)$ with weights $\boldsymbol{\lambda}^*$, then $R_T = regret_T(\boldsymbol{\lambda}^*)$.*

*Proof.* Since $(\boldsymbol{x}^*, \boldsymbol{\lambda}^*)$ is the optimal solution of problem (6), we then have $R_T = \min_{\boldsymbol{\lambda} \in \Lambda} \max_{\boldsymbol{x} \in X} \boldsymbol{\lambda}^T \sum_{t=1}^{T} (\boldsymbol{F}_t(\boldsymbol{x}_t) - \boldsymbol{F}_t(\boldsymbol{x})) = \max_{\boldsymbol{x} \in X} \boldsymbol{\lambda}^{*T} \sum_{t=1}^{T} (\boldsymbol{F}_t(\boldsymbol{x}_t) - \boldsymbol{F}_t(\boldsymbol{x})) = \boldsymbol{\lambda}^{*T} \sum_{t=1}^{T} \boldsymbol{F}_t(\boldsymbol{x}_t) - \min_{\boldsymbol{x} \in X} \boldsymbol{\lambda}^{*T} \sum_{t=1}^{T} \boldsymbol{F}_t(\boldsymbol{x}) = regret_T(\boldsymbol{\lambda}^*)$. $\square$

**Proposition 2.** *For an OMCO problem, which is transformed into an OCO problem $(P_\lambda)$ with weights $\bar{\boldsymbol{\lambda}} \in \Lambda$, if an OCO algorithm achieves a sublinear bound for the weighted sum regret $regret_T(\bar{\boldsymbol{\lambda}})$, then this algorithm also guarantees a sublinear multi-objective regret in the OMCO framework.*

*Proof.* Let $(\boldsymbol{x}^*, \boldsymbol{\lambda}^*)$ be an optimal solution of problem (6), from Proposition 1, we have $R_T = \boldsymbol{\lambda}^{*T} \sum_{t=1}^T \boldsymbol{F}_t(\boldsymbol{x}_t) - \min_{\boldsymbol{x} \in X} \boldsymbol{\lambda}^{*T} \sum_{t=1}^T \boldsymbol{F}_t(\boldsymbol{x}) \leqslant \bar{\boldsymbol{\lambda}}^T \sum_{t=1}^T \boldsymbol{F}_t(\boldsymbol{x}_t) - \min_{\boldsymbol{x} \in X} \bar{\boldsymbol{\lambda}}^T \sum_{t=1}^T \boldsymbol{F}_t(\boldsymbol{x}) = regret_T(\bar{\boldsymbol{\lambda}})$. $\qquad\square$

Similarly, it is easy to establish a multi-objective regret with respect to a time-varying comparator sequence in dynamic settings (see Section C for details).

## 5 ONLINE JOINT GRADIENT DESCENT ALGORITHM

In this section, we try to minimize the worst-case objective value and design an online joint gradient descent algorithm for OMCO problems, as well as provide its a multi-objective regret bound.

### 5.1 ALGORITHM DEVELOPMENT

To resolve potential conflicts among objectives in problem (4) from a global perspective, we use a robustness-driven method that tries to minimize the worst-case objective value as follows.

$$\min_{\boldsymbol{x} \in X} \max\{\sum_{t=1}^T F_t^1(\boldsymbol{x}), \cdots, \sum_{t=1}^T F_t^p(\boldsymbol{x})\}. \tag{8}$$

**Lemma 2.** *An optimal solution $\boldsymbol{x}^*$ of the problem (8) is a weakly efficient solution of problem (4).*

*Proof.* We prove Lemma 2 by contradiction. Since $\boldsymbol{x}^*$ is an optimal solution of problem (8), then for all $\boldsymbol{x} \in X$, we have $\max_{i \in [p]} \sum_{t=1}^T F_t^i(\boldsymbol{x}^*) \leqslant \max_{i \in [p]} \sum_{t=1}^T F_t^i(\boldsymbol{x})$. Suppose $\boldsymbol{x}^*$ is not a weakly efficient solution of the problem (4), then there exists another solution $\bar{\boldsymbol{x}} \in X$ such that $\sum_{t=1}^T F_t^i(\bar{\boldsymbol{x}}) < \sum_{t=1}^T F_t^i(\boldsymbol{x}^*)$ for all $i$. This means that $\max_{i \in [p]} \sum_{t=1}^T F_t^i(\bar{\boldsymbol{x}}) < \max_{i \in [p]} \sum_{t=1}^T F_t^i(\boldsymbol{x}^*)$, which contradicts with $\max_{i \in [p]} \sum_{t=1}^T F_t^i(\boldsymbol{x}^*) \leqslant \max_{i \in [p]} \sum_{t=1}^T F_t^i(\boldsymbol{x})$. Therefore, Lemma 2 holds. $\qquad\square$

To better facilitate the design of OMCO algorithm, we introduce weight factors to transform problem (8) into the following form.

$$\min_{\boldsymbol{x} \in X} \max_{\boldsymbol{\lambda} \in \Lambda} \boldsymbol{\lambda}^T \sum_{t=1}^T \boldsymbol{F}_t(\boldsymbol{x}). \tag{9}$$

We further establish the relationship between the problems (8) and (9) according to the following lemma, the proof of which is provided in Section D.

**Lemma 3.** *An optimal solution of the problem (9) is an optimal solution of the problem (8).*

According to Lemma 2 and 3, by solving problem (9), we can obtain a solution that performs well for the worst-case in the multi-objective trade-off. As a result, this solution is ensured to be weakly non-dominated and robust over the entire feasible domain in the objective space. For this minimax problem (9), let its instantaneous optimization function be $\mathcal{L}_t(\boldsymbol{x}, \boldsymbol{\lambda}) := \boldsymbol{\lambda}^T \boldsymbol{F}_t(\boldsymbol{x})$. Then we introduce the online joint gradient descent algorithm, which takes a descent step to update the decision $\boldsymbol{x} \in X$ and an ascent step to update the weights $\boldsymbol{\lambda} \in \Lambda$ at each $t$-th round, i.e.,

$$\boldsymbol{x}_{t+1} = \Pi_X(\boldsymbol{x}_t - \eta_t^p \nabla_x \mathcal{L}_t(\boldsymbol{x}_t, \boldsymbol{\lambda}_t)), \quad \boldsymbol{\lambda}_{t+1} = \Pi_\Lambda(\boldsymbol{\lambda}_t + \eta_t^d (\nabla_\lambda \mathcal{L}_t(\boldsymbol{x}_t, \boldsymbol{\lambda}_t) + \alpha_t \Delta(\boldsymbol{\lambda}_t))),$$

where $\nabla_x \mathcal{L}_t$ and $\nabla_\lambda \mathcal{L}_t$ are the gradients of $\mathcal{L}_t$ with respect to $\boldsymbol{x}$ and $\boldsymbol{\lambda}$, respectively. $\Delta(\boldsymbol{\lambda}_t) := \frac{\boldsymbol{\lambda}_1 - \boldsymbol{\lambda}_t}{\|\boldsymbol{\lambda}_1 - \boldsymbol{\lambda}_t\|}$ is a regularization term used to avoid the weights drifting over time. $\eta_t^p$ and $\eta_t^d$ are the decision stepsize and weights stepsize, respectively. $\alpha_t$ is a trade-off factor that is used to balance exploration and convergence. In fact, letting $\nabla \boldsymbol{F}_t(\boldsymbol{x})$ be the Jacobian matrix of $\boldsymbol{F}_t(\boldsymbol{x})$, then $\nabla_x \mathcal{L}_t(\boldsymbol{x}_t, \boldsymbol{\lambda}_t) = \nabla \boldsymbol{F}_t(\boldsymbol{x}_t)^T \boldsymbol{\lambda}_t$ is a joint gradient and all objective functions decrease along the joint gradient direction. In addition, the decision stepsize is actually the learning rate in applications. The complete algorithm is summarized in Algorithm 1.

---

**Algorithm 1** Online Joint Gradient Descent (OJGD)

---

**Require:** $X, T$, learning rates $\{\eta_t^p\}$, weights stepsizes $\{\eta_t^d\}$, $\boldsymbol{x}_1 \in X$, $\boldsymbol{\lambda}_1 \in \Lambda$.
1: **for** $t = 1$ to $T$ **do**
2:    Predict $\boldsymbol{x}_t$ and then receive $\boldsymbol{F}_t$
3:    Suffer loss $\boldsymbol{F}_t(\boldsymbol{x}_t)$, and observe $\nabla \boldsymbol{F}_t(\boldsymbol{x}_t)$
4:    Update the decisions:

$$\boldsymbol{x}_{t+1} = \Pi_X(\boldsymbol{x}_t - \eta_t^p \nabla \boldsymbol{F}_t(\boldsymbol{x}_t)^T \boldsymbol{\lambda}_t) \tag{10}$$

5:    Update the weights: $\Delta(\boldsymbol{\lambda}_t) = \boldsymbol{0}$ if $\boldsymbol{\lambda}_t = \boldsymbol{\lambda}_1$ and $\Delta(\boldsymbol{\lambda}_t) = \frac{\boldsymbol{\lambda}_1 - \boldsymbol{\lambda}_t}{\|\boldsymbol{\lambda}_1 - \boldsymbol{\lambda}_t\|}$ otherwise, then

$$\boldsymbol{\lambda}_{t+1} = \Pi_\Lambda(\boldsymbol{\lambda}_t + \eta_t^d(\boldsymbol{F}_t(\boldsymbol{x}_t) + \alpha_t \Delta(\boldsymbol{\lambda}_t))) \tag{11}$$

6: **end for**

---

It's worth noting that the projection operation for the update of weights in Algorithm 1 is a projection onto a simplex, it is straightforward to implement in linear time (Duchi et al., 2008). Therefore, the OJGD algorithm is easier to implement and takes less time than the methods based on minimizing (regularized) the joint gradient (Jiang et al., 2023; Désidéri, 2012; Sener & Koltun, 2018), especially in complex problems, which is verified in Section 6.

## 5.2 PERFORMANCE ANALYSIS

We provide performance analyses of Algorithm 1 based on two basic assumptions (Hazan, 2022).

**Assumption 1.** *The domain $X \subset \mathbb{R}^n$ is convex and bounded, i.e., $\max_{\boldsymbol{x}_1, \boldsymbol{x}_2 \in X} \|\boldsymbol{x}_1 - \boldsymbol{x}_2\| \leqslant D$.*

**Assumption 2.** *The gradients of all loss functions are convex and bounded by a constant $G$, i.e., $\forall t \in [T], i \in [p], \|\nabla F_t^i(\boldsymbol{x})\| \leqslant G$.*

**Theorem 3.** *Suppose Assumptions 1 and 2 hold, assume $F_t^i$ is bounded, i.e., $\forall \boldsymbol{x} \in X, t \in [T], i \in [p], |F_t^i(\boldsymbol{x})| \leqslant F$, set the trade-off factor $\alpha_t = Ft^\tau$ and the directional gain factor $\tau \geqslant 0$, the OJGD attains the following multi-objective regret*

$$R_T \leqslant \frac{D^2}{2\eta_T^p} + \frac{G^2}{2}\sum_{t=1}^T \eta_t^p + \frac{\sqrt{2}}{2\eta_T^d} + 2pF^2\sum_{t=1}^T \eta_t^d t^{2\tau}.$$

**Remark 2.** *When stepsizes $\eta_t^p = \frac{D}{G\sqrt{t}}$, $\eta_t^d = \frac{1}{F\sqrt{t}}$ and $\tau \in [0, 0.25)$, the OJGD achieve a subliner multi-objective regret bound. The proof of Theorem 3 is provided in Section D. Moreover, we provide the regret analysis of OJGD under the strongly convex and dynamic settings in Sections D.*

**Dynamic phase properties of $\alpha_t$.** According to the setting of $\alpha_t$ in Theorem 3, the dynamic phase properties of the trade-off factor $\alpha_t$ have an adaptive adjustment mechanism. Specifically, for the optimization direction $(\boldsymbol{F}_t(\boldsymbol{x}_t) + \alpha_t \Delta(\boldsymbol{\lambda}_t))$ of $\boldsymbol{\lambda}_{t+1}$ in (11), the first term $\boldsymbol{F}_t(\boldsymbol{x}_t)$ reflects the performance of each objective at $t$-th round, the second term $\Delta(\boldsymbol{\lambda}_t)$ reflects the orientation of weights $\boldsymbol{\lambda}_t$ to weights $\boldsymbol{\lambda}_1$. Since $\alpha_t = Ft^\tau$ and $\tau \in [0, 0.25)$, in the early stages of the iteration, OJGD improves the underperforming objectives while the optimization direction is also influenced by the initial weights, i.e., it is in the exploration stage. As the iterative process evolves, the weights of the joint gradient are guided by the initial weights to ensure the asymptotic stability of the (weakly) efficient solution, i.e., OJGD gradually transitions to the convergence stage.

## 6 EXPERIMENTS

In this section, we validate the effectiveness of our algorithm on classical convex machine learning models and verify its generalizability on non-convex models. We conduct experiments to compare OJGD with three baselines: (i) uniform scaling: linearization performs single-objective online learning on scalar losses with equal weights; (ii) min-norm: obtaining the joint gradient via min-norm of gradient composition (Désidéri, 2012; Sener & Koltun, 2018); (iii) DR-OMMD: using weights for the gradient composition via min-regularized-norm (Jiang et al., 2023). For each experiment, initial weights are set as equal weights. More experimental results and details are deferred to Section E.

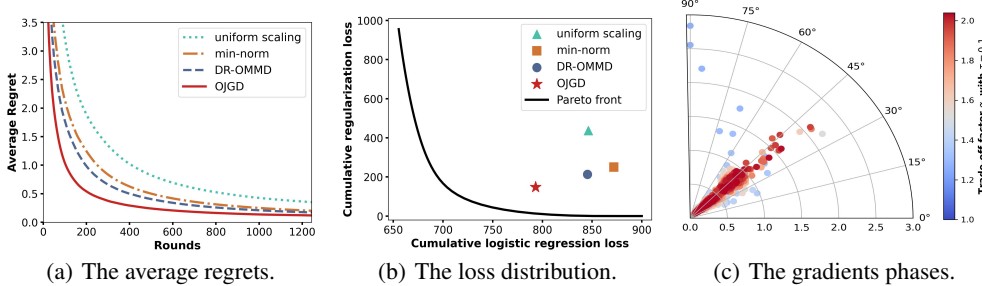

(a) The average regrets.  (b) The loss distribution.  (c) The gradients phases.

Figure 1: The results on the svmguide3 dataset. Figure 1(a) plots the time-average multi-objective regrets of all the algorithms; Figure 1(b) shows the distribution of cumulative loss values for the algorithms in the objective space and the Pareto front of the OMCO problem; the phases of the joint gradients of the OJGD algorithm during the optimization process are plotted in Figure 1(c).

## 6.1 CONVEX EXPERIMENTS

Linear regression and logistic regression have well-studied convex objectives. In machine learning, the loss function usually combines an empirical loss and a regularization term, which often conflict. Specifically, minimizing empirical loss may lead to overfitting, while strong regularization may lead to underfitting of the data. Thus, model training can be formulated as a multi-objective optimization problem, treating the empirical loss and regularization as two separate objectives that need to be balanced. We use **mg** and **svmguide3** datasets from the UCI Machine Learning repository (Dua et al., 2017). For the prediction/classification problem, we use linear regression/logistic regression loss as the first objective function and squared the L2-norm of the model parameters as the second objective function. We restrict the model parameters to a sphere $X$ with a radius $R = 10$.

In Figure 1(a), we plot the time-average regret $R_T/T$ curve on the svmguide3 dataset. The results show that OJGD achieves the lowest regret among all the algorithms. Figure 1(b) plots the cumulative losses obtained by the algorithms and the Pareto fronts of the OMCO probelms at $T$-th round, where the Pareto fronts are obtained by the RNBI method (Shao & Ehrgott, 2016). This figure intuitively shows that the cumulative loss obtained by OJGD is closer to the Pareto front than those obtained by other algorithms and reflects that OJGD can better trade off the two objectives.

Furthermore, we show the phase of the joint gradients under the variation of the trade-off factor $\alpha_t$ during the optimization process in Figure 1(c). Specifically, for the $t$-th round, the phase $\theta_t$ is obtained from the joint gradient weights via $\theta_t = \arctan 2(\boldsymbol{\lambda}_t^2, \boldsymbol{\lambda}_t^1)$, and the radius $r_t$ is the norm of the joint gradient, i.e., $r_t = \|\nabla \boldsymbol{F}_t(\boldsymbol{x}_t)^T \boldsymbol{\lambda}_t\|$. The colors of the points in Figure 1(c) are determined by the size of $\alpha_t$, reflecting the optimization progress. From Figure 1(c), OJGD explores various optimization directions in the early stage. As the trade-off factor $\alpha_t$ increases, the phase of the joint gradient is guided by the initial weight direction and is gradually pulled back to the initial weight direction, and the norm of the joint gradient gradually becomes smaller. The above variation process of the joint gradient phase reflects the evolution of OJGD from the exploration stage to the convergence stage.

## 6.2 NON-CONVEX EXPERIMENTS

We use standard multi-task supervised learning datasets: MultiMNIST (Sabour et al., 2017) and NYU-v2 (Silberman et al., 2012). The MultiMNIST dataset contains 2 tasks: simultaneously classifying the digit on the top-left (task-L) and the bottom-right (task-R) separately. The NYU-v2 dataset contains 3 tasks: 13-class semantic segmentation, depth estimation, and surface normal estimation. For MultiMNIST, we follow the setup of Sener & Koltun (2018) and use the LeNet network architecture. For NYU-v2, we run all experiments using the MTL benchmark framework LibMTL (Lin & Zhang, 2023) with the MTAN (Liu et al., 2019) network architecture.

For MultiMNIST, Figure 2 shows the distribution of the average cumulative losses obtained by the algorithms in the objective space and accuracy profile in detecting the left and right digits for the algorithms. Specifically, the train losses are plotted in Figure 2(a), the test losses are plotted in

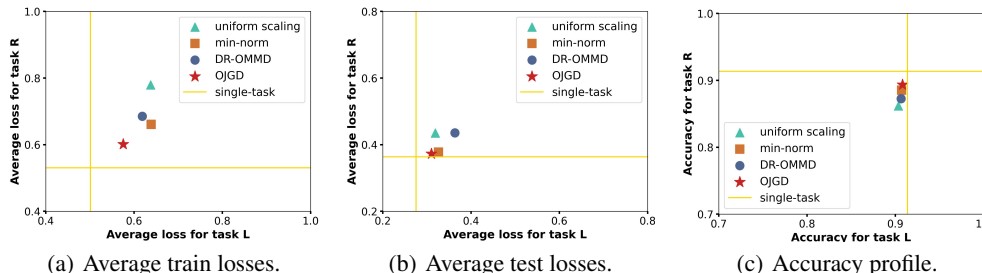

(a) Average train losses.  (b) Average test losses.  (c) Accuracy profile.

Figure 2: The results on the MultiMNIST dataset. Figure 2(a) and 2(b) plot the distribution of the average cumulative train losses and test losses in the objective space of all the algorithms; Figure 2(c) shows the distribution of accuracy in detecting the left and right digits for all the algorithms.

Figure 2(b), and the accuracy profile is shown in Figure 2(c). Since it is difficult to obtain the Pareto front of this non-convex problem model, we consider constructing the ideal point of this problem to compare the algorithms' results more clearly, where the ideal point consists of the optimal value of each objective. We use the **single-task** (i.e., online solving tasks independently) as an added baseline. The vector consisting of the single-task's results can be approximated as an ideal point, i.e., the intersection of the two yellow lines in Figure 2. From Figure 2, OJGD's results are both closer to the approximate ideal points constituted by the single-task's results than the results of the other algorithms.

For NYU-v2, we follow the evaluation metrics in Liu et al. (2019) for the three tasks. Moreover, we follow the work of Maninis et al. (2019) and report the overall performance metric $\Delta M\%$ by computing the average per-task performance drop of an algorithm $a$ versus the single-task baseline $b$. Specifically, $\Delta M\% = \frac{1}{N_m}\sum_{n=1}^{N_m}(-1)^{l_n}(m_{a,n}-m_{b,n})/m_{b,n} \times 100$, where $N_m$ is the number of metrics, $m_{a,n}$ and $m_{b,n}$ are respectively the $n$-th metric values for algorithms $a$ and single-task $b$, $l_n = 1$ if higher $n$-th metric is better and 0 otherwise. In addition, to measure the performance efficiency per unit time of each algorithm, we report the average running time $\Delta t$ of each algorithm for a single epoch and the Performance-Time (P-T) ratio, i.e., $\frac{1/\Delta M\%}{\Delta t} \times 100$. We report the above metrics in Table 1, which shows that OJGD obtains better or competitive results than other algorithms. Furthermore, OJGD has the best value of P-T ratio, which means the overall performance is the best over the same time.

Table 1: The results on the NYU-v2 dataset. The mean of each metric (↑: higher better; ↓: lower better) is reported over 3 independent runs. The best average result is marked in bold.

| Method | Segmentation | | Depth | | Surface Normal | | | | | | $\Delta M\%\downarrow$ | $\Delta t(s)\downarrow$ | P-T ↑ |
| --- | --- | --- | --- | --- | --- | --- | --- | --- | --- | --- | --- | --- | --- |
| | | | | | Angle Distance ↓ | | Within $t°$ ↑ | | | | | | |
| | mIoU ↑ | Pix Acc ↑ | Abs Err ↓ | Rel Err ↓ | Mean | Median | 11.25 | 22.5 | 30 | | | | |
| single-task | 52.89 | 74.69 | 0.4185 | 0.1706 | 21.81 | 14.72 | 40.03 | 65.95 | 75.74 | | | |
| uniform scaling | 52.88 | 74.79 | 0.3848 | 0.1565 | 22.98 | 16.32 | 36.46 | 62.46 | 73.20 | 1.93 | **73.18** | 0.7351 |
| min-norm | 45.87 | 69.56 | 0.4172 | 0.1707 | **21.86** | **15.07** | **39.18** | **65.24** | **75.39** | 2.92 | 278.93 | 0.1247 |
| DR-OMMD | 51.57 | 73.85 | 0.3932 | 0.1623 | 22.47 | 15.76 | 37.55 | 63.79 | 74.62 | 1.53 | 281.45 | 0.2339 |
| **OJGD** | **53.11** | **75.04** | **0.3812** | **0.1552** | 22.98 | 16.39 | 37.12 | 63.01 | 73.47 | **1.41** | 78.26 | **0.9130** |

## 7 CONCLUSIONS

In this paper, we extended the OCO framework to multi-objective cases and established an OMCO framework, which unifies the online convex optimization framework for any number of objectives. Additionally, we derived an equivalent form of multi-objective regret. Then, we proposed an Online Joint Gradient Descent (OJGD), which is proved to guarantee a sublinear regret bound. Finally, numerical experiments consistently demonstrate that OJGD outperforms the other algorithms across multiple real-world datasets. While our analysis is mainly based on convex objectives, an interesting direction is to consider non-convex objectives under an online multi-objective optimization setup.

ETHICS STATEMENT

The authors declare that they have no known competing financial interests or personal relationships that could have appeared to influence the work reported in this paper.

REPRODUCIBILITY STATEMENT

The implementation of the algorithm proposed in this paper and all experiments are reproducible. We have uploaded the anonymized source code to the supplementary materials. The assumptions and proofs of the lemmas and theorems are detailed in Section 5 and D. The dataset sources and experimental settings are fully described in Section 6 and E.

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

# Online Multi-objective Convex Optimization: A Unified Framework and Joint Gradient Descent (Supplementary material)

This supplementary material is organized as follows:

- **Appendix A** introduces more related works.
- **Appendix B** analyzes why multi-objective regret under the domain of real numbers matters.
- **Appendix C** further discusses multi-objective regret of OMCO in a dynamic environment.
- **Appendix D** supplements the proofs omitted for the OJGD algorithm in the main paper.
- **Appendix E** provides more experimental details and results.
- **Appendix F** discloses the use of Large Language Models (LLMs).

## A  More related works

Since our OJGD algorithm is designed based on the minimax problem (9), some intersection works between online optimization methods and minimax problems are also related to our work. There have been some efforts to study the OCO methods to solve minimax problems with time-invariant objective functions (Orabona, 2019; Gupta et al., 2020; Bampis et al., 2022). Moreover, Lee et al. (2022) studied the online minimax problem with vector losses. In the setting of Lee et al. (2022), the learner makes a decision after the loss functions are revealed.

However, our work is largely different from these works in the following aspects. In our OMCO framework, the objective functions (i.e., loss functions) change with time, which is different from the settings in the works of Orabona (2019); Gupta et al. (2020); Bampis et al. (2022). Furthermore, in the setting of the OMCO framework, the loss functions in each round are revealed only after the decision has been made, which is different from the setting in Lee et al. (2022). Therefore, the methods in all the above works cannot handle the minimax problem (9) under the OMCO framework.

## B  Why the multi-objective regret under the domain of real numbers matters

In this section, we will present why the multi-objective regret under the domain of real numbers matters. Specifically, this work has both theoretical value and practical significance. Our decision to extend regret to $\mathbb{R}$ is motivated by three key considerations:

**Mathematical Completeness:** Prior work (Jiang et al., 2023) restricts a multi-objective regret to $\mathrm{Reg}_T \geqslant 0$, which introduces a discontinuity between the single-objective setting $\mathrm{Reg}_T \in \mathbb{R}$ and multi-objective regimes. Our unified regret framework resolves this discrepancy (formally proven in Theorem 1) and establishes a more complete theoretical foundation for future algorithm design.

**Theoretical Possibility:** As mentioned in Section 1, it is theoretically possible to observe negative regret when the algorithm's decisions $x_t$ happen to coincide with the instantaneous optimal decision of an objective. For instance, in our new synthetic experiments on a two-objective linear regression task (i.e., the following problem (12)), the uniform scaling algorithm indeed achieves a convergent negative regret bound, illustrating that negative values can occur under specific conditions.

**Practical Guidance:** From our synthetic experiments for problem (12), achieving negative regret correlates strongly with three factors: the problem structure, sample distribution, and the algorithm's update strategy. These findings suggest that negative regret is not just a theoretical curiosity, but provides practical insights guiding algorithm design tailored to specific problem instances.

Next, we report the setup and results of this synthetic experiment for problem (12).

**Negative Regret Experiment Setup and Results:** We consider an online two-objective optimization problem:

$$\min_{x \in [0,10]} \left( \sum_{t=1}^{T} (a_t x - b_t)^2, \ \sum_{t=1}^{T} x^2 \right), \quad T = 200, \tag{12}$$

where all data pairs $\forall t \in [T], (a_t, b_t) \in \mathbb{R} \times \mathbb{R}$ are generated before running the experiment. We apply the uniform scaling algorithm with step size $\eta_t = 0.08/\sqrt{t}$, initializing $x_1 = 1$. The following Figure 3 displays the average regret and the distribution of the average cumulative losses in the objective space obtained by the uniform scaling algorithm.

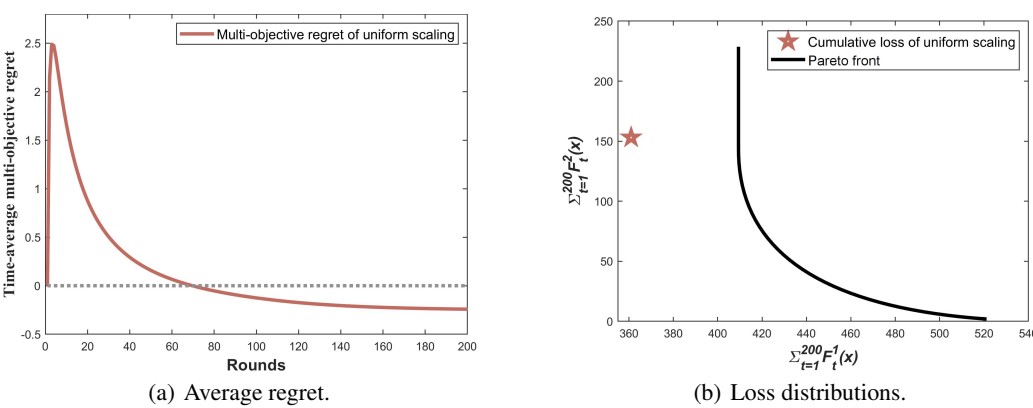

(a) Average regret.        (b) Loss distributions.

Figure 3: The average regret and the distribution of the average cumulative losses in the objective space obtained by the uniform scaling algorithm.

In this experiment, negative regret occurs because at multiple times, the algorithm's decision $x_t$ exactly matches the optimal decision of the first objective at the following time. Thus, on average, the incurred loss is lower than that of the hindsight best fixed decision for that objective.

This finding supports our theoretical claim that negative regret is possible under specific conditions, and demonstrates how our regret framework extending into $\mathbb{R}$ can capture such phenomena.

Given the above reasons, relaxing the non-negativity constraint on multi-objective regret is both a significant theoretical contribution and offers practical value.

## C DYNAMIC REGRET IN OMCO

The multi-objective regret we proposed in the main paper is analyzed based on a fixed comparator in a static setting, which is also called static regret. Next, we analyze the dynamic regret with respect to time-varying comparators in the OMCO framework. We first review single-objective cases, the dynamic regret in OCO is defined with respect to a time-varying comparator sequence as follows.

$$regret_T^D = \sum_{t=1}^{T} f_t(\boldsymbol{x}_t) - \sum_{t=1}^{T} \min_{\boldsymbol{y}_t \in X} f_t(\boldsymbol{y}_t). \tag{13}$$

In dynamic settings, for all $t \in [T]$, we set $\boldsymbol{y}_t^* = arg\min_{\boldsymbol{y}_t \in X} f_t(\boldsymbol{y}_t)$. Then, the regret bounds of algorithms are often related to the path-length of the minimizer sequence $\{\boldsymbol{y}_t^*\}$, i.e., $L_* = \sum_{t=2}^{T} \|\boldsymbol{y}_t^* - \boldsymbol{y}_{t-1}^*\|$. For example, OGD enjoys an $O(L_*\sqrt{T})$ regret for convex functions (Zinkevich, 2003) and an $O(L_*)$ regret for strongly convex and smooth convex functions (Mokhtari et al., 2016). Now we continue our study of the OMCO framework for dynamic situations.

**Definition 5.** *The dynamic regret in multi-objective cases is*

$$R_T^D = \sum_{t=1}^{T} \max_{\boldsymbol{y}_t \in X, \delta_t \in \mathbb{R}} \delta_t, \ s.t. \ \boldsymbol{F}_t(\boldsymbol{x}_t) - \delta_t \boldsymbol{e} \geqslant \boldsymbol{F}_t(\boldsymbol{y}_t), \forall t \in [T]. \tag{14}$$

Similar to static regret, we have the following theorems and propositions.

**Theorem 4.** *When the number of objectives is degenerated to one, $R_T^D$ is equal to the dynamic regret in single-objective case, i.e.,*

$$R_T^D = regret_T^D, \ when \ p = 1.$$

*Proof.* When $p = 1$, we have

$$R_T^D = \sum_{t=1}^{T} \max_{\boldsymbol{y}_t \in X, \delta_t \in \mathbb{R}} \{\delta_t, \ s.t. f_t(\boldsymbol{x}_t) - \delta_t \geqslant f_t(\boldsymbol{y}_t), t \in [T]\}$$

$$= \sum_{t=1}^{T} \max_{\boldsymbol{y}_t \in X} (f_t(\boldsymbol{x}_t) - f_t(\boldsymbol{y}_t))$$

$$= \sum_{t=1}^{T} f_t(\boldsymbol{x}_t) - \sum_{t=1}^{T} \min_{\boldsymbol{y}_t \in X} f_t(\boldsymbol{y}_t) = regret_T^D.$$

Therefore, Theorem 4 holds. $\square$

**Remark 3.** *Similarly, we can use '$R_T^D$' to denote the dynamic regret of an optimization algorithm for an online convex optimization problem with any number of objectives.*

**Theorem 5.** *For each $t \in [T]$, assume that there exists $\delta_t^0 \in \mathbb{R}$ and $\boldsymbol{x}_0 \in intX$ such that $\boldsymbol{F}_t(\boldsymbol{x}_t) - \delta_t^0 \boldsymbol{e} > \boldsymbol{F}_t(\boldsymbol{x}_0)$ holds, then an equivalent form of the dynamic online multi-objective regret is*

$$R_T^D = \sum_{t=1}^{T} \min_{\boldsymbol{\lambda}_t \in \Lambda} \max_{\boldsymbol{y}_t \in X} \boldsymbol{\lambda}_t^T (\boldsymbol{F}_t(\boldsymbol{x}_t) - \boldsymbol{F}_t(\boldsymbol{y}_t)). \tag{15}$$

*Proof.* The problem (14) is a convex optimization problem and its Lagrangian dual problem is as follows,

$$\sum_{t=1}^{T} \min_{\boldsymbol{\lambda}_t \in \mathbb{R}_{\geqslant}} \max_{\boldsymbol{y}_t \in X, \delta_t \in \mathbb{R}} \delta_t + \boldsymbol{\lambda}_t^T (\boldsymbol{F}_t(\boldsymbol{x}_t) - \delta_t \boldsymbol{e} - \boldsymbol{F}_t(\boldsymbol{y}_t)), \tag{16}$$

where $\boldsymbol{\lambda}_t$ is the dual variable.

According to the assumption, it is known that Slater's condition is satisfied. Thus, the problem pair (14) and (16) has strong duality, i.e., their optimal values are equal. Further, by the KKT conditions, the derivative of objective function of problem (16) with respect to $\delta_t$ is zero, we have $\boldsymbol{\lambda}_t \in \Lambda$. Then, problem (16) is equivalent to the right-hand side of (15). Therefore, Theorem 5 holds. $\square$

Next, we introduce a weighted sum method under dynamic settings to give the relationship between our proposed dynamic regret and weighted sum method. We consider a single-objective optimization problem with time-varying weights $\boldsymbol{\lambda}_t \in \Lambda, \forall t \in [T]$ as follows.

$$\min \sum_{t=1}^{T} \boldsymbol{\lambda}_t^T \boldsymbol{F}_t(\boldsymbol{y}_t), \ \boldsymbol{y}_t \in X \text{ and } \forall t \in [T]. \tag{17}$$

The the weighted sum dynamic regret is

$$regret_T^D(\boldsymbol{\lambda}_t) = \sum_{t=1}^{T} \boldsymbol{\lambda}_t^T \boldsymbol{F}_t(\boldsymbol{x}_t) - \sum_{t=1}^{T} \min_{\boldsymbol{y}_t \in X} \boldsymbol{\lambda}_t^T \boldsymbol{F}_t(\boldsymbol{y}_t). \tag{18}$$

**Proposition 3.** *Suppose that $(\boldsymbol{y}_t^*, \boldsymbol{\lambda}_t^*)$ is an optimal solution of the above problem (15), if an OMCO problem is transformed into a weighted sum OCO problem (17) with weights $\boldsymbol{\lambda}_t^*$, then*

$$R_T^D = regret_T^D(\boldsymbol{\lambda}_t^*). \tag{19}$$

*Proof.* Since $(\boldsymbol{y}_t^*, \boldsymbol{\lambda}_t^*)$ is the optimal solution of problem (15) for $t$-th round, then we have

$$
\begin{aligned}
R_T^D &= \sum_{t=1}^{T} \min_{\boldsymbol{\lambda}_t \in \Lambda} \max_{\boldsymbol{y}_t \in X} \boldsymbol{\lambda}_t^T (\boldsymbol{F}_t(\boldsymbol{x}_t) - \boldsymbol{F}_t(\boldsymbol{x})) \\
&= \sum_{t=1}^{T} \max_{\boldsymbol{y}_t \in X} \boldsymbol{\lambda}_t^{*T} (\boldsymbol{F}_t(\boldsymbol{x}_t) - \boldsymbol{F}_t(\boldsymbol{y}_t)) \\
&= \sum_{t=1}^{T} \boldsymbol{\lambda}_t^{*T} \boldsymbol{F}_t(\boldsymbol{x}_t) - \sum_{t=1}^{T} \min_{\boldsymbol{y}_t \in X} \boldsymbol{\lambda}_t^{*T} \boldsymbol{F}_t(\boldsymbol{y}_t) \\
&= regret_T^D(\boldsymbol{\lambda}_t^*).
\end{aligned}
$$

$\square$

**Proposition 4.** *For an OMCO problem, which is transformed into an OCO problem (17) with weights $\bar{\boldsymbol{\lambda}}_t \in \Lambda$, if an OCO algorithm achieve sublinear bound for the weighted sum dynamic regret (18), then the OCO algorithm can also achieve sublinear dynamic regret in the OMCO framework.*

*Proof.* Let $(\boldsymbol{y}_t^*, \boldsymbol{\lambda}_t^*)$ is an optimal solution of problem (15), from Proposition 3, we have

$$
\begin{aligned}
R_T^D &= \sum_{t=1}^{T} \boldsymbol{\lambda}_t^{*T} \boldsymbol{F}_t(\boldsymbol{x}_t) - \sum_{t=1}^{T} \min_{\boldsymbol{y}_t \in X} \boldsymbol{\lambda}_t^{*T} \boldsymbol{F}_t(\boldsymbol{y}_t) \\
&\leqslant \sum_{t=1}^{T} \bar{\boldsymbol{\lambda}}_t^T \boldsymbol{F}_t(\boldsymbol{x}_t) - \sum_{t=1}^{T} \min_{\boldsymbol{y}_t \in X} \bar{\boldsymbol{\lambda}}_t^T \boldsymbol{F}_t(\boldsymbol{y}_t) \\
&= regret_T(\bar{\boldsymbol{\lambda}}).
\end{aligned}
$$

$\square$

# D   OMITTED PROOFS FOR THE OJGD ALGORITHM

## D.1   PROOF OF LEMMA 3

Before proving Lemma 3 1, we introduce the following lemma.

**Lemma 4.** *The problems (8) and (9) have the following relationship: for all $\boldsymbol{x} \in X$,*

$$\max\{\sum_{t=1}^{T} F_t^1(\boldsymbol{x}), \cdots, \sum_{t=1}^{T} F_t^p(\boldsymbol{x})\} = \max_{\boldsymbol{\lambda} \in \Lambda} \boldsymbol{\lambda}^T \sum_{t=1}^{T} \boldsymbol{F}_t(\boldsymbol{x}).$$

*Proof.* Suppose that the problem $\max_{i \in [p]} \sum_{t=1}^{T} F_t^i(\boldsymbol{x})$ achieves the optimal value at $i = j \in [p]$, i.e., $\max\{\sum_{t=1}^{T} F_t^1(\boldsymbol{x}), \cdots, \sum_{t=1}^{T} F_t^p(\boldsymbol{x})\} = \sum_{t=1}^{T} F_t^j(\boldsymbol{x})$. Then for all $\boldsymbol{\lambda} \in \Lambda$, we have $\boldsymbol{\lambda}^T \sum_{t=1}^{T} \boldsymbol{F}_t(\boldsymbol{x}) - \sum_{t=1}^{T} F_t^j(\boldsymbol{x}) = \boldsymbol{\lambda}^T (\sum_{t=1}^{T} \boldsymbol{F}_t(\boldsymbol{x}) - \boldsymbol{e} \cdot \sum_{t=1}^{T} F_t^j(\boldsymbol{x})) \leqslant 0$. Therefore, $\forall \boldsymbol{\lambda} \in \Lambda$, $\boldsymbol{\lambda}^T \sum_{t=1}^{T} \boldsymbol{F}_t(\boldsymbol{x}) \leqslant \sum_{t=1}^{T} F_t^j(\boldsymbol{x})$. This implies that $\max_{\boldsymbol{\lambda} \in \Lambda} \boldsymbol{\lambda}^T \sum_{t=1}^{T} \boldsymbol{F}_t(\boldsymbol{x}) = \sum_{t=1}^{T} F_t^j(\boldsymbol{x})$, where an optimal solution $\boldsymbol{\lambda}^* \in \Lambda$ could be taken as $\boldsymbol{\lambda} = \boldsymbol{e}_j$. Consequently, Lemma 4 holds. $\square$

The proof of Lemma 3 is as follows.

*Proof.* Let $(\boldsymbol{x}^*, \boldsymbol{\lambda}^*)$ be an optimal solution of the problem (9). Based on Lemma 4, $\forall \boldsymbol{x} \in X$, we have

$$\max_{i \in [p]}\{\sum_{t=1}^{T} F_t^i(\boldsymbol{x})\} = \max_{\boldsymbol{\lambda} \in \Lambda} \boldsymbol{\lambda}^T \sum_{t=1}^{T} \boldsymbol{F}_t(\boldsymbol{x}) \geqslant \min_{\boldsymbol{x} \in X} \max_{\boldsymbol{\lambda} \in \Lambda} \boldsymbol{\lambda}^T \sum_{t=1}^{T} \boldsymbol{F}_t(\boldsymbol{x}) = \boldsymbol{\lambda}^{*T} \sum_{t=1}^{T} \boldsymbol{F}_t(\boldsymbol{x}^*).$$

According to Lemma 4, we have $\boldsymbol{\lambda}^{*T} \sum_{t=1}^{T} \boldsymbol{F}_t(\boldsymbol{x}^*) = \max\{\sum_{t=1}^{T} F_t^1(\boldsymbol{x}^*), \cdots, \sum_{t=1}^{T} F_t^p(\boldsymbol{x}^*)\}$ and can take $\boldsymbol{x} = \boldsymbol{x}^*$ to obtain the problem (8)'s optimal value. Thus, Lemma 3 holds. $\qquad\square$

### D.2 PROOF OF THEOREM 3

The proof of Theorem 3 is as follows.

*Proof.* Let $\boldsymbol{x}^*$ be an optimal solution of problem (6) and considring Theorem 2, we have

$$\begin{aligned}
R_T \leqslant & \boldsymbol{\lambda}_1^T (\sum_{t=1}^{T} \boldsymbol{F}_t(\boldsymbol{x}_t) - \sum_{t=1}^{T} \boldsymbol{F}_t(\boldsymbol{x}^*)) \\
= & \underbrace{\sum_{t=1}^{T} \boldsymbol{\lambda}_t^T (\boldsymbol{F}_t(\boldsymbol{x}_t) - \boldsymbol{F}_t(\boldsymbol{x}^*))}_{\text{Part 1}} + \underbrace{\sum_{t=1}^{T} (\boldsymbol{\lambda}_1 - \boldsymbol{\lambda}_t)^T (\boldsymbol{F}_t(\boldsymbol{x}_t) - \boldsymbol{F}_t(\boldsymbol{x}^*))}_{\text{Part 2}}.
\end{aligned} \quad (20)$$

For Part 1 in (20), by using the update rule for $\boldsymbol{x}_{t+1}$ in (10), we have

$$\|\boldsymbol{x}_{t+1} - \boldsymbol{x}^*\|^2 \leqslant \|\boldsymbol{x}_t - \boldsymbol{x}^*\|^2 + \eta_t^{p\,2} \|\nabla \boldsymbol{F}_t(\boldsymbol{x}_t)^T \boldsymbol{\lambda}_t\|^2 - 2\eta_t^p \boldsymbol{\lambda}_t^T \nabla \boldsymbol{F}_t(\boldsymbol{x}_t)(\boldsymbol{x}_t - \boldsymbol{x}^*).$$

Rearranging the above terms, it follows that

$$2\boldsymbol{\lambda}_t^T \nabla \boldsymbol{F}_t(\boldsymbol{x}_t)(\boldsymbol{x}_t - \boldsymbol{x}^*) \leqslant \frac{\|\boldsymbol{x}_t - \boldsymbol{x}^*\|^2 - \|\boldsymbol{x}_{t+1} - \boldsymbol{x}^*\|^2}{\eta_t^p} + \eta_t^p \|\nabla \boldsymbol{F}_t(\boldsymbol{x}_t)^T \boldsymbol{\lambda}_t\|^2.$$

We define $\frac{1}{\eta_0^p} := 0$ and assume the learning rate $\eta_t^p$ is non-increasing over time. Using the convexity and summing the above terms from $t = 1$ to $T$, then

$$\begin{aligned}
2\sum_{t=1}^{T} \boldsymbol{\lambda}_t (\boldsymbol{F}_t(\boldsymbol{x}_t) - \boldsymbol{F}_t(\boldsymbol{x}^*)) \leqslant & 2\sum_{t=1}^{T} \boldsymbol{\lambda}_t^T \nabla \boldsymbol{F}_t(\boldsymbol{x}_t)(\boldsymbol{x}_t - \boldsymbol{x}^*) \\
\leqslant & \sum_{t=1}^{T} \|\boldsymbol{x}_t - \boldsymbol{x}^*\|^2 (\frac{1}{\eta_t^p} - \frac{1}{\eta_{t-1}^p}) + G^2 \sum_{t=1}^{T} \eta_t^p \\
\leqslant & D^2 \frac{1}{\eta_T^p} + G^2 \sum_{t=1}^{T} \eta_t^p
\end{aligned} \quad (21)$$

For Part 2 in (20), by using the update rule for $\boldsymbol{\lambda}_{t+1}$ in (11), we have

$$\begin{aligned}
& \|\boldsymbol{\lambda}_{t+1} - \boldsymbol{\lambda}_1\|^2 \\
= & \|\Pi_\Lambda(\boldsymbol{\lambda}_t + \eta_t^d (\boldsymbol{F}_t(\boldsymbol{x}_t) + \alpha_t \Delta(\boldsymbol{\lambda}_t))) - \boldsymbol{\lambda}_1\|^2 \\
\leqslant & \|\boldsymbol{\lambda}_t + \eta_t^d (\boldsymbol{F}_t(\boldsymbol{x}_t) + \alpha_t \Delta(\boldsymbol{\lambda}_t)) - \boldsymbol{\lambda}_1\|^2 \\
= & \|\boldsymbol{\lambda}_t - \boldsymbol{\lambda}_1\|^2 + 2\eta_t^d (\boldsymbol{\lambda}_t - \boldsymbol{\lambda}_1)^T (\boldsymbol{F}_t(\boldsymbol{x}_t) + \alpha_t \Delta(\boldsymbol{\lambda}_t)) + (\eta_t^d)^2 \|\boldsymbol{F}_t(\boldsymbol{x}_t) + \alpha_t \Delta(\boldsymbol{\lambda}_t)\|^2
\end{aligned}$$

Rearranging the above terms, it follows that

$$2(\boldsymbol{\lambda}_1 - \boldsymbol{\lambda}_t)^T \boldsymbol{F}_t(\boldsymbol{x}_t) \leqslant \frac{\|\boldsymbol{\lambda}_t - \boldsymbol{\lambda}_1\|^2 - \|\boldsymbol{\lambda}_{t+1} - \boldsymbol{\lambda}_1\|^2}{\eta_t^d} + \eta_t^d p(F + \alpha_t)^2 - 2\alpha_t (\boldsymbol{\lambda}_1 - \boldsymbol{\lambda}_t)^T \Delta(\boldsymbol{\lambda}_t)$$

Both sides of the above inequality are subtracted from $2(\boldsymbol{\lambda}_1 - \boldsymbol{\lambda}_t)^T \boldsymbol{F}_t(\boldsymbol{x}^*)$ at the same time, We then consider the following two cases.

**1) case 1:** when $\boldsymbol{\lambda}_t = \boldsymbol{\lambda}_1$, we have

$$2(\boldsymbol{\lambda}_1 - \boldsymbol{\lambda}_t)^T(\boldsymbol{F}_t(\boldsymbol{x}_t) - \boldsymbol{F}_t(\boldsymbol{x}^*))$$

$$\leqslant \frac{\|\boldsymbol{\lambda}_t - \boldsymbol{\lambda}_1\|^2 - \|\boldsymbol{\lambda}_{t+1} - \boldsymbol{\lambda}_1\|^2}{\eta_t^d} + \eta_t^d p(F + \alpha_t)^2 - 2\alpha_t(\boldsymbol{\lambda}_1 - \boldsymbol{\lambda}_t)^T\Delta(\boldsymbol{\lambda}_t) - 2(\boldsymbol{\lambda}_1 - \boldsymbol{\lambda}_t)^T\boldsymbol{F}_t(\boldsymbol{x}^*)$$

$$= \frac{\|\boldsymbol{\lambda}_t - \boldsymbol{\lambda}_1\|^2 - \|\boldsymbol{\lambda}_{t+1} - \boldsymbol{\lambda}_1\|^2}{\eta_t^d} + \eta_t^d p(F + \alpha_t)^2$$

**2) case 2:** when $\boldsymbol{\lambda}_t \neq \boldsymbol{\lambda}_1$, set $\alpha_t = Ft^\tau$, where $\tau \geqslant 0$, then $\Delta(\boldsymbol{\lambda}_t) = \frac{\boldsymbol{\lambda}_1 - \boldsymbol{\lambda}_t}{\|\boldsymbol{\lambda}_1 - \boldsymbol{\lambda}_t\|}$ and we have

$$2(\boldsymbol{\lambda}_1 - \boldsymbol{\lambda}_t)^T(\boldsymbol{F}_t(\boldsymbol{x}_t) - \boldsymbol{F}_t(\boldsymbol{x}^*))$$

$$\leqslant \frac{\|\boldsymbol{\lambda}_t - \boldsymbol{\lambda}_1\|^2 - \|\boldsymbol{\lambda}_{t+1} - \boldsymbol{\lambda}_1\|^2}{\eta_t^d} + \eta_t^d p(F + \alpha_t)^2 - 2\alpha_t(\boldsymbol{\lambda}_1 - \boldsymbol{\lambda}_t)^T\Delta(\boldsymbol{\lambda}_t) - 2(\boldsymbol{\lambda}_1 - \boldsymbol{\lambda}_t)^T\boldsymbol{F}_t(\boldsymbol{x}^*)$$

$$\leqslant \frac{\|\boldsymbol{\lambda}_t - \boldsymbol{\lambda}_1\|^2 - \|\boldsymbol{\lambda}_{t+1} - \boldsymbol{\lambda}_1\|^2}{\eta_t^d} + \eta_t^d p(F + \alpha_t)^2 - 2\alpha_t\|\boldsymbol{\lambda}_1 - \boldsymbol{\lambda}_t\| + 2F\|\boldsymbol{\lambda}_1 - \boldsymbol{\lambda}_t\|$$

$$= \frac{\|\boldsymbol{\lambda}_t - \boldsymbol{\lambda}_1\|^2 - \|\boldsymbol{\lambda}_{t+1} - \boldsymbol{\lambda}_1\|^2}{\eta_t^d} + \eta_t^d p(F + \alpha_t)^2 + 2\|\boldsymbol{\lambda}_1 - \boldsymbol{\lambda}_t\|(F - \alpha_t)$$

$$\leqslant \frac{\|\boldsymbol{\lambda}_t - \boldsymbol{\lambda}_1\|^2 - \|\boldsymbol{\lambda}_{t+1} - \boldsymbol{\lambda}_1\|^2}{\eta_t^d} + \eta_t^d p(F + \alpha_t)^2$$

Thus, combining cases 1 and 2, we have

$$2(\boldsymbol{\lambda}_1 - \boldsymbol{\lambda}_t)^T(\boldsymbol{F}_t(\boldsymbol{x}_t) - \boldsymbol{F}_t(\boldsymbol{x}^*)) \leqslant \frac{\|\boldsymbol{\lambda}_t - \boldsymbol{\lambda}_1\|^2 - \|\boldsymbol{\lambda}_{t+1} - \boldsymbol{\lambda}_1\|^2}{\eta_t^d} + \eta_t^d p(F + \alpha_t)^2.$$

Then, summing the above terms from $t = 1$ to $T$ and set $\alpha_t = Ft^\tau$, where $\tau \geqslant 0$, then we can obtain an upper bound of Part 2 in (20) as follows.

$$2\sum_{t=1}^T (\boldsymbol{\lambda}_1 - \boldsymbol{\lambda}_t)^T(\boldsymbol{F}_t(\boldsymbol{x}_t) - \boldsymbol{F}_t(\boldsymbol{x}^*)) \leqslant \sum_{t=1}^T \frac{\|\boldsymbol{\lambda}_t - \boldsymbol{\lambda}_1\|^2 - \|\boldsymbol{\lambda}_{t+1} - \boldsymbol{\lambda}_1\|^2}{\eta_t^d} + \eta_t^d p(F + \alpha_t)^2$$

$$\leqslant \sum_{t=2}^T \|\boldsymbol{\lambda}_t - \boldsymbol{\lambda}_1\|^2 (\frac{1}{\eta_t^d} - \frac{1}{\eta_{t-1}^d}) + 4pF^2 \sum_{t=1}^T \eta_t^d t^{2\tau}$$

$$\leqslant \sqrt{2}\frac{1}{\eta_T^d} + 4pF^2 \sum_{t=1}^T \eta_t^d t^{2\tau} \qquad (22)$$

Finally, according to (21) and (22), we obtain an upper bound on the regret of OJGD for convex functions as shown in Theorem 3.

$\square$

### D.3 REGRET ANALYSIS IN THE STRONGLY CONVEX SETTING

In this section, we analyze the multi-objective regret bound of the OJGD algorithm in the strongly convex setting, where the strongly convex function is defined as follows.

**Definition 6.** *A function $f$ is $H$-strongly convex over the convex set $X$ if $f(\boldsymbol{x}_1) \geqslant f(\boldsymbol{x}_2) + \nabla f(\boldsymbol{x}_2)^T(\boldsymbol{x}_1 - \boldsymbol{x}_2) + \frac{H}{2}\|\boldsymbol{x}_1 - \boldsymbol{x}_2\|^2, \forall \boldsymbol{x}_1, \boldsymbol{x}_2 \in X.$*

**Theorem 6.** *Suppose Assumptions 1 and 2 hold, assume all functions are $H$-strongly convex functions and $F_t^i$ is bounded, i.e., $\forall \boldsymbol{x} \in X, t \in [T], i \in [p], |F_t^i(\boldsymbol{x})| \leqslant F$, let stepsizes $\eta_t^p = \frac{1}{Ht}$ and $\eta_t^d = \frac{1}{F\sqrt{t}}$, set $\alpha_t = Ft^\tau$ and $\tau \in [0, 0.25)$, the OJGD attains the following multi-objective regret*

$$R_T \leqslant \frac{G^2}{H}(1 + \log T) + F\sqrt{2T} + \frac{4pF}{2\tau + 0.5}T^{2\tau + 0.5}.$$

*Proof.* Let $\boldsymbol{x}^*$ be an optimal solution of problem (5). From Theorem 2, we have

$$R_T = \min_{\boldsymbol{\lambda} \in \Lambda} \max_{\boldsymbol{x} \in X} \boldsymbol{\lambda}^T (\sum_{t=1}^T \boldsymbol{F}_t(\boldsymbol{x}_t) - \sum_{t=1}^T \boldsymbol{F}_t(\boldsymbol{x}))$$

$$\leqslant \boldsymbol{\lambda}_1^T (\sum_{t=1}^T \boldsymbol{F}_t(\boldsymbol{x}_t) - \sum_{t=1}^T \boldsymbol{F}_t(\boldsymbol{x}^*))$$

$$= \underbrace{\sum_{t=1}^T \boldsymbol{\lambda}_t^T (\boldsymbol{F}_t(\boldsymbol{x}_t) - \boldsymbol{F}_t(\boldsymbol{x}^*))}_{\text{Part 1}} + \underbrace{\sum_{t=1}^T (\boldsymbol{\lambda}_1 - \boldsymbol{\lambda}_t)^T (\boldsymbol{F}_t(\boldsymbol{x}_t) - \boldsymbol{F}_t(\boldsymbol{x}^*))}_{\text{Part 2}}. \tag{23}$$

For the above Part 1 and 2 in (23), we further separately prove the upper bounds. For Part 1, based on the update rule for $\boldsymbol{x}_{t+1}$, we have

$$\|\boldsymbol{x}_{t+1} - \boldsymbol{x}^*\|^2 = \|\Pi_X(\boldsymbol{x}_t - \eta_t^p \nabla \boldsymbol{F}_t(\boldsymbol{x}_t)^T \boldsymbol{\lambda}_t) - \boldsymbol{x}^*\|^2$$

$$\leqslant \|\boldsymbol{x}_t - \eta_t^p \nabla \boldsymbol{F}_t(\boldsymbol{x}_t)^T \boldsymbol{\lambda}_t - \boldsymbol{x}^*\|^2$$

$$\leqslant \|\boldsymbol{x}_t - \boldsymbol{x}^*\|^2 + \eta_t^{p2} \|\nabla \boldsymbol{F}_t(\boldsymbol{x}_t)^T \boldsymbol{\lambda}_t\|^2 - 2\eta_t^p \boldsymbol{\lambda}_t^T \nabla \boldsymbol{F}_t(\boldsymbol{x}_t)(\boldsymbol{x}_t - \boldsymbol{x}^*).$$

Rearranging the above terms, it follows that

$$2\boldsymbol{\lambda}_t^T \nabla \boldsymbol{F}_t(\boldsymbol{x}_t)(\boldsymbol{x}_t - \boldsymbol{x}^*) \leqslant \frac{\|\boldsymbol{x}_t - \boldsymbol{x}^*\|^2 - \|\boldsymbol{x}_{t+1} - \boldsymbol{x}^*\|^2}{\eta_t^p} + \eta_t^p \|\nabla \boldsymbol{F}_t(\boldsymbol{x}_t)^T \boldsymbol{\lambda}_t\|^2.$$

Let $\eta_t^p = \frac{1}{Ht}$, by the $H$-strong convexity of the loss functions, we have

$$2\sum_{t=1}^T \boldsymbol{\lambda}_t^T (\boldsymbol{F}_t(\boldsymbol{x}_t) - \boldsymbol{F}_t(\boldsymbol{x}^*)) \leqslant 2\sum_{t=1}^T \boldsymbol{\lambda}_t^T \nabla \boldsymbol{F}_t(\boldsymbol{x}_t)(\boldsymbol{x}_t - \boldsymbol{x}^*) - \sum_{t=1}^T H\|\boldsymbol{x}_t - \boldsymbol{x}^*\|^2$$

$$\leqslant \sum_{t=1}^T \|\boldsymbol{x}_t - \boldsymbol{x}^*\|^2 (\frac{1}{\eta_t^p} - \frac{1}{\eta_{t-1}^p} - H) + G^2 \sum_{t=1}^T \eta_t^p$$

$$\leqslant 0 + G^2 \sum_{t=1}^T \frac{1}{Ht}$$

$$\leqslant \frac{G^2}{H}(1 + \log T). \tag{24}$$

Let $\eta_t^p = \frac{1}{F\sqrt{t}}$, $\alpha_t = Ft^\tau$ and $\tau \in [0, 0.25)$, the analyze Part 2 in (23) is similar to that of Part 2 of (20) in the proof of Theorem 3, we have

$$2\sum_{t=1}^T (\boldsymbol{\lambda}_1 - \boldsymbol{\lambda}_t)^T (\boldsymbol{F}_t(\boldsymbol{x}_t) - \boldsymbol{F}_t(\boldsymbol{x}^*)) \leqslant \sqrt{2}\frac{1}{\eta_T^d} + 4pF^2 \sum_{t=1}^T \eta_t^d t^{2\tau} \leqslant F\sqrt{2T} + \frac{4pF}{2\tau + 0.5}T^{2\tau + 0.5}. \tag{25}$$

Finally, we summarize the above results. According to (24) and (25), the Theorem 3 holds.

$\square$

## D.4 DYNAMIC REGRET ANALYSIS FOR OJGD

In this section, based on the dynamic multi-objective regret in (14), we analyze the dynamic multi-objective regret bound of the OJGD algorithm.

**Theorem 7.** *Suppose Assumptions 1 and 2 hold and all functions are convex. If $\eta_t^p = \eta := \frac{\sqrt{D(D+2L_*)}}{G\sqrt{T}}$, where $L_* = \sum_{t=2}^T \|\boldsymbol{y}_{t-1}^* - \boldsymbol{y}_t^*\|$ and $\boldsymbol{y}_t^*$ is an optimal solution of problem (14) at $t$-th round, the OJGD attains the following dynamic regret*

$$R_T^D \leqslant G\sqrt{D(D + 2L_*)T}.$$

*Proof.* Based on the update rule for $\boldsymbol{x}_{t+1}$, we have

$$
\begin{aligned}
\|\boldsymbol{x}_{t+1} - \boldsymbol{y}_t^*\|^2 &= \|\Pi_X(\boldsymbol{x}_t - \eta \nabla \boldsymbol{F}_t(\boldsymbol{x}_t)^T \boldsymbol{\lambda}_t) - \boldsymbol{y}_t^*\|^2 \\
&\leqslant \|\boldsymbol{x}_t - \eta \nabla \boldsymbol{F}_t(\boldsymbol{x}_t)^T \boldsymbol{\lambda}_t - \boldsymbol{y}_t^*\|^2 \\
&\leqslant \|\boldsymbol{x}_t - \boldsymbol{y}_t^*\|^2 + \eta^2 \|\nabla \boldsymbol{F}_t(\boldsymbol{x}_t)^T \boldsymbol{\lambda}_t\|^2 - 2\eta \boldsymbol{\lambda}_t^T \nabla \boldsymbol{F}_t(\boldsymbol{x}_t)(\boldsymbol{x}_t - \boldsymbol{y}_t^*).
\end{aligned}
$$

Thus,

$$
2\boldsymbol{\lambda}_t^T \nabla \boldsymbol{F}_t(\boldsymbol{x}_t)(\boldsymbol{x}_t - \boldsymbol{y}_t^*) \leqslant \frac{\|\boldsymbol{x}_t - \boldsymbol{y}_t^*\|^2 - \|\boldsymbol{x}_{t+1} - \boldsymbol{y}_t^*\|^2}{\eta} + \eta \|\nabla \boldsymbol{F}_t(\boldsymbol{x}_t)^T \boldsymbol{\lambda}_t\|^2.
$$

Summing the above inequality from $t = 1$ to $T$, and considering the convexity of the loss functions and Theorem 5, we have

$$
\begin{aligned}
2R_T^D &= 2\sum_{t=1}^{T} \min_{\bar{\boldsymbol{\lambda}}_t \in \Lambda} \max_{\boldsymbol{y}_t \in X} \bar{\boldsymbol{\lambda}}_t^T (\boldsymbol{F}_t(\boldsymbol{x}_t) - \boldsymbol{F}_t(\boldsymbol{y}_t)) \\
&\leqslant 2\sum_{t=1}^{T} \boldsymbol{\lambda}_t^T (\boldsymbol{F}_t(\boldsymbol{x}_t) - \boldsymbol{F}_t(\boldsymbol{y}_t^*)) \\
&\leqslant 2\sum_{t=1}^{T} \boldsymbol{\lambda}_t^T \nabla \boldsymbol{F}_t(\boldsymbol{x}_t)(\boldsymbol{x}_t - \boldsymbol{y}_t^*) \\
&\leqslant \sum_{t=1}^{T} \frac{\|\boldsymbol{x}_t - \boldsymbol{y}_t^*\|^2 - \|\boldsymbol{x}_{t+1} - \boldsymbol{y}_t^*\|^2}{\eta} + \sum_{t=1}^{T} \eta \|\nabla \boldsymbol{F}_t(\boldsymbol{x}_t)^T \boldsymbol{\lambda}_t\|^2 \\
&\leqslant \frac{1}{\eta}\|\boldsymbol{x}_1 - \boldsymbol{y}_1^*\|^2 + \frac{1}{\eta}\sum_{t=2}^{T}(\|\boldsymbol{x}_t - \boldsymbol{y}_t^*\|^2 - \|\boldsymbol{x}_t - \boldsymbol{y}_{t-1}^*\|^2) - \frac{1}{\eta}\|\boldsymbol{x}_{T+1} - \boldsymbol{y}_T^*\|^2 + \eta G^2 T \\
&\leqslant \frac{D^2}{\eta} + \frac{1}{\eta}\sum_{t=2}^{T} \|\boldsymbol{x}_t - \boldsymbol{y}_t^* + \boldsymbol{x}_t - \boldsymbol{y}_{t-1}^*\|\|\boldsymbol{y}_{t-1}^* - \boldsymbol{y}_t^*\| + \eta G^2 T \\
&\leqslant \frac{D^2}{\eta} + \frac{1}{\eta}\sum_{t=2}^{T} 2D\|\boldsymbol{y}_{t-1}^* - \boldsymbol{y}_t^*\| + \eta G^2 T \\
&= \frac{D(D + 2L_*)}{\eta} + \eta G^2 T.
\end{aligned}
$$

Set $\eta := \frac{\sqrt{D(D + 2L_*)}}{G\sqrt{T}}$, then Theorem 7 holds. $\qquad\square$

# E    MORE EXPERIMENTAL RESULTS

In this section, we provide more details of the experiment and the experimental results. All experiments are conducted on a single NVIDIA GeForce RTX 4090 GPU, PyTorch 1.8.1 platform.

## E.1    MORE RESULTS FOR CONVEX EXPERIMENTS

The **mg** and **svmguide3** datasets used in our experiments are publicly available in Dua et al. (2017). The mg dataset contains 1385 instances with 6 features and the svmguide3 dataset contains 1243 instances with 21 features. For the two convex problems, the experiments are conducted in the online setting, i.e., the samples arrive sequentially. For the mg dataset, the directional gain factor $\tau$ in $\alpha_t = Ft^\tau$ of OJGD is set as $\tau = 0.2$. For the svmguide3 dataset, the directional gain factor $\tau$ is set as $\tau = 0.1$. We supplement the results in Figure 4 and Figure 5.

Figure 4 show that the OJGD algorithm has the best overall performance. Remarkably, the min-norm method obtains the lowest average loss in the regularization loss, however, it performs the worst on

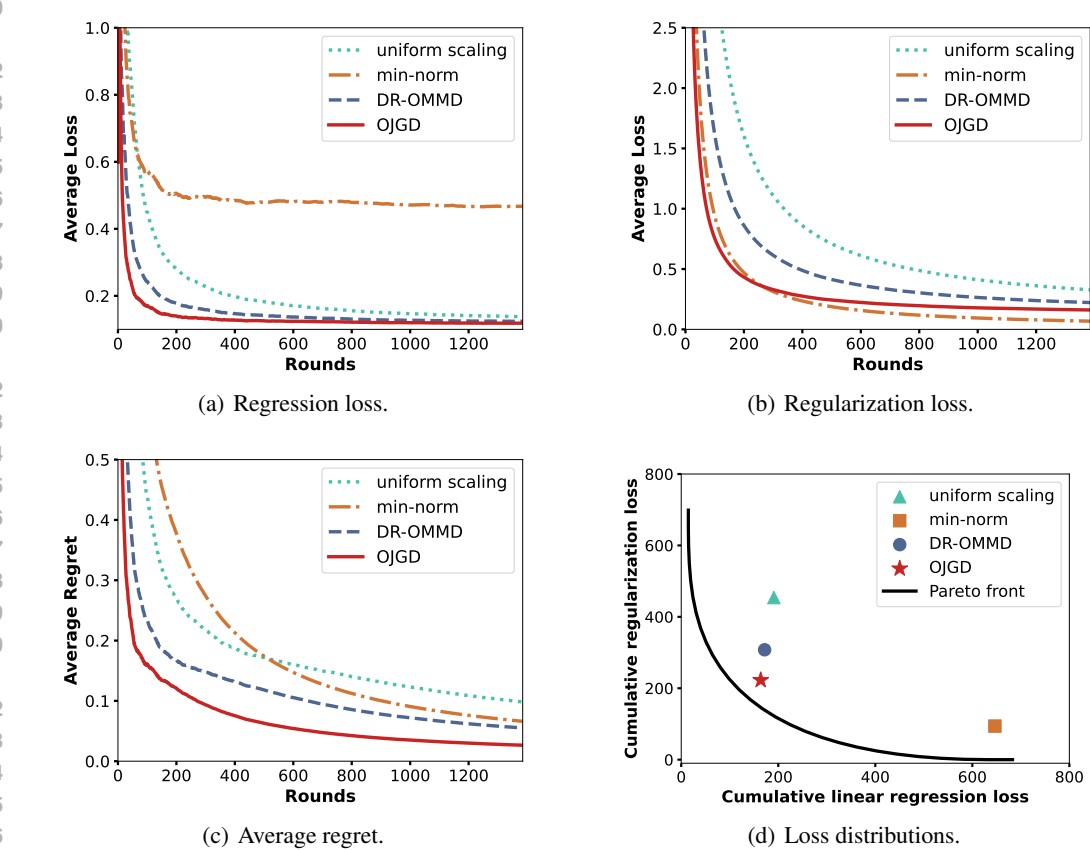

(a) Regression loss.

(b) Regularization loss.

(c) Average regret.

(d) Loss distributions.

Figure 4: More empirical results of the algorithms on the mg dataset. The plots show the average cumulative losses in (a) and (b). The average multi-objective regrets are shown in (c). The distributions of cumulative losses are shown in (d).

the regression objective. This may be because the min-norm method updates the weights only based on minimizing the instantaneous joint gradient and lacks a regularization strategy for the weights, which may change drastically following the gradient, making it difficult to trade off multiple objectives in the optimization process stably. Figure 5 shows that OJGD consistently outperforms other algorithms on the svmguide3 dataset.

Furthermore, we explore the effect of different directional gain factors $\tau$ on the gradient phase. We set $\tau \in \{0, 0.05, 0.1, 0.15, 0.2, 0.24\}$ and implement the OJGD algorithm with the same settings as in the above experiments. The results on the svmguide3 dataset are shown in Figure 6. Specifically, for the $t$-th round, the phase $\theta_t$ is obtained from the joint gradient weights via $\theta_t = \arctan 2(\boldsymbol{\lambda}_t^2, \boldsymbol{\lambda}_t^1)$, and the radius $r_t$ is the norm of the joint gradient, i.e., $r_t = \|\nabla \boldsymbol{F}_t(\boldsymbol{x}_t)^T \boldsymbol{\lambda}_t\|$. The colors of the points in Figure 6 are determined by the size of $\alpha_t$, reflecting the optimization progress.

The results in Figure 6 show that even when $\tau = 0$, the OJGD algorithm still has some exploratory ability to adaptively adjust the weights during the optimization process. Compared with the performance of OJGD when $\tau = 0$, the OJGD algorithm with $\tau \in \{0.05, 0.1\}$ exhibits a similar exploratory ability in the early stage, but its convergence ability in the later stage is significantly enhanced, i.e., the weights stabilize around the initial weights in the later stage. It is worth noting that the OJGD algorithm's exploratory ability in the medium stages is further enhanced when $\tau \in \{0.15, 0.2, 0.24\}$. This is because when the directional gain factor $\tau$ becomes large, the trade-off factor $\alpha_t$ also becomes large quickly. In the weight update direction $(\boldsymbol{F}_t(\boldsymbol{x}_t) + \alpha_t \Delta(\boldsymbol{\lambda}_t))$, the second term $\alpha_t \Delta(\boldsymbol{\lambda}_t)$ quickly becomes the dominant factor compared to the first term $\boldsymbol{F}_t(\boldsymbol{x}_t)$. In this case, the force in the algorithm to pull the weights back to the initial weights is too large, making the algorithm have exploration capabilities in the medium stage. As the stepsize $\eta_t^d$ becomes gradually smaller, this pulling force also becomes smaller, allowing the algorithm to converge gradually. We

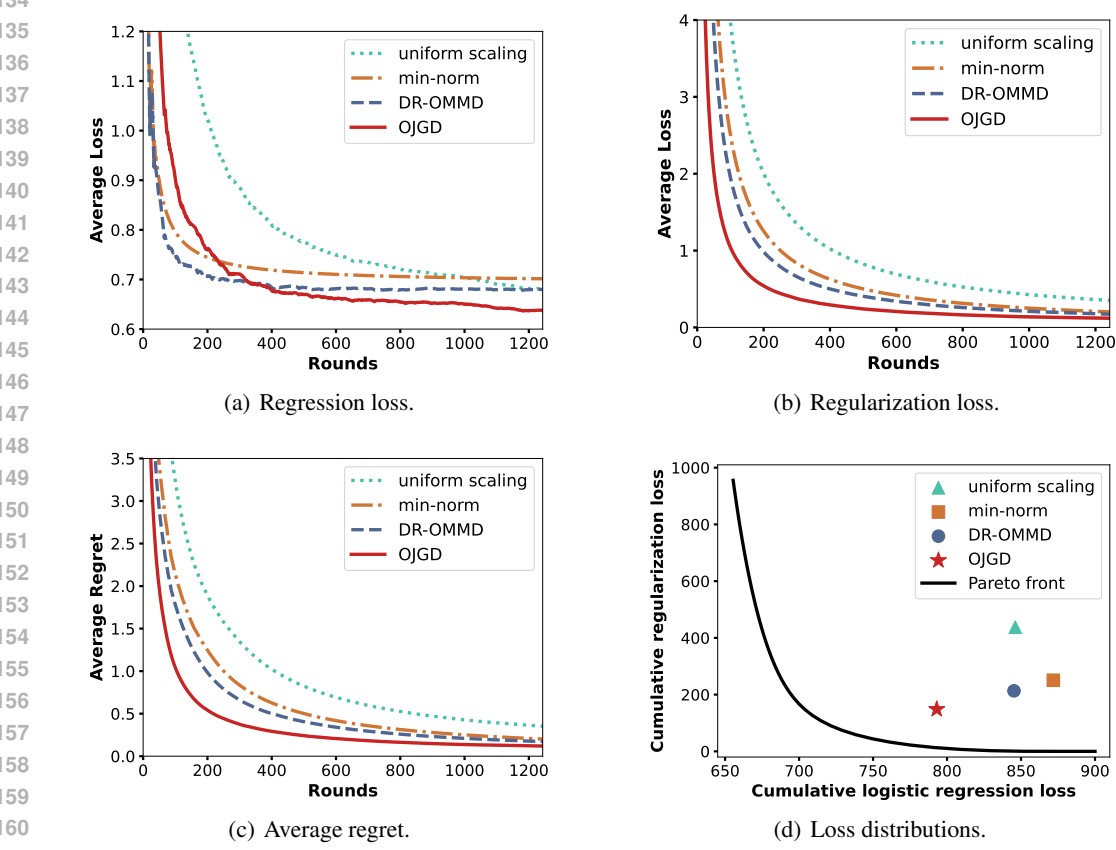

(a) Regression loss.

(b) Regularization loss.

(c) Average regret.

(d) Loss distributions.

Figure 5: More empirical results of the algorithms on the svmguide3 dataset. The plots show the average cumulative losses in (a) and (b). The average multi-objective regrets are shown in (c). The distributions of cumulative losses are shown in (d).

further plot the distribution of the obtained cumulative losses in Figure 7, showing that the OJGD algorithm with $\tau = 0.2$ performs best.

### E.2 MORE RESULTS FOR NON-CONVEX EXPERIMENTS

The MultiMNIST dataset is acquired by the code provided by Sener & Koltun (2018). It is a multi-task version of the MNIST dataset, where each sample consists of an overlap of two digital images from MNIST at the top-left and another one at the bottom-right. The tasks are simultaneously classifying the digit on the top-left (task-L) and the bottom-right (task-R) separately. Following the experimental setup of Jiang et al. (2023), this experiments are also conducted in the online setting and the learning rates of the algorithms are decided by a grid search over $\{0.0001, 0.001, 0.01, 0.1\}$. The stepsizes $\eta_t^d$ and $\alpha_t$ of OJGD is set according to Algorithm 1. We supplement the results of the algorithms on MultiMNIST in Figure 8. The results show that OJGD consistently outperforms other algorithms.

The NYU-v2 dataset (Silberman et al., 2012) consists of images from indoor video sequences, including image segmentation, depth estimation, and surface normal estimation tasks. We follow the evaluation metrics in Liu et al. (2019) for the three tasks. Specifically, for semantic segmentation, we use Mean Intersection over Union (mIoU) and Pixel Accuracy (Pix Acc). For depth estimation, we use Absolute Error (Abs Err) and Relative Error (Rel Err). For surface normal estimation, we use five metrics, i.e., mean absolute of the error (Mean), median absolute of the error (Median), and the percentages of pixels with the angular error below $11.25°$, $22.5°$, and $30°$ (denoted as 11.25, 22.5, 30). In addition, following Maninis et al. (2019), we also compute the following metrics to reflect the overall performance of an algorithm $a$: the average per-task performance drop versus the

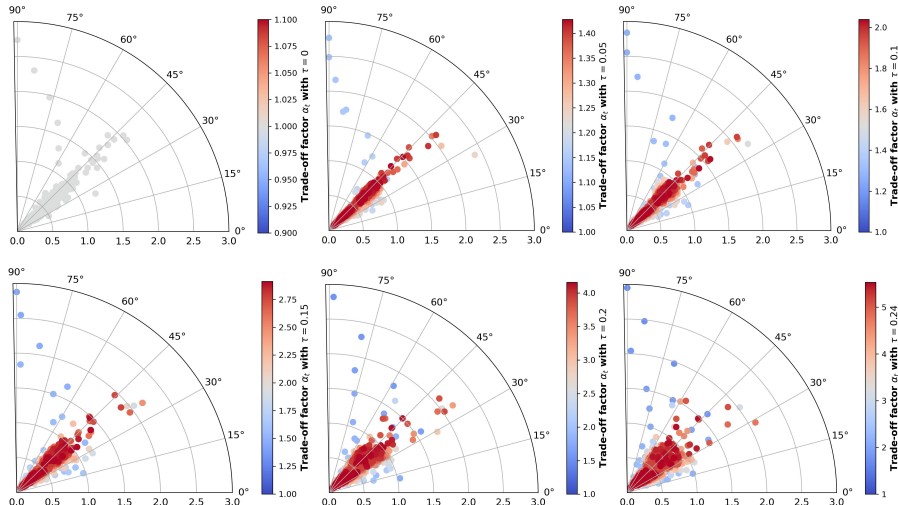

Figure 6: The phases of the joint gradients of the OJGD algorithm with different directional gain factors $\tau \in \{0, 0.05, 0.1, 0.15, 0.2, 0.24\}$ during the optimization process on the svmguide3 dataset.

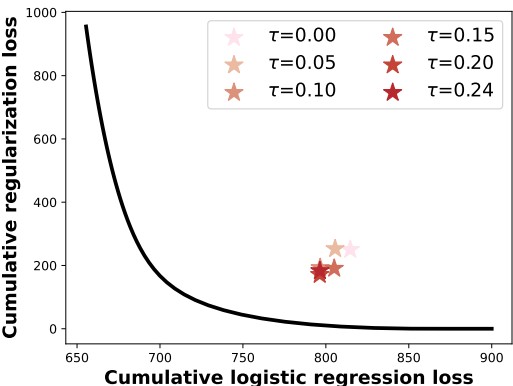

Figure 7: The distribution of the cumulative losses obtained by OJGD on svmguide3.

single-task baseline $b$, i.e., $\Delta M\% = \frac{1}{N_m} \sum_{n=1}^{N_m} (-1)^{l_n} (m_{a,n} - m_{b,n})/m_{b,n} \times 100$, where $m_{a,n}$ and $m_{b,n}$ are respectively the $n$-th metric values for algorithms $a$ and single-task $b$, $l_n = 1$ if higher $n$-th metric is better and $0$ otherwise. We run all experiments using the MTL benchmark framework LibMTL (Lin & Zhang, 2023) with the MTAN (Liu et al., 2019) network architecture. The stepsizes $\eta_t^d$ and $\alpha_t$ of OJGD is set according to Algorithm 1. For all algorithms, we trained MTAN with 200 epochs and a batch size of 2, and the default LibMTL configuration is used for all other experiment setups as shown in Table 2.

Table 2: Summary of hyper-parameter choices for NYU-v2 dataset.

| Method | optimizer of $x_t$ | learning rate | weights stepsize | weight decay |
|---|---|---|---|---|
| uniform scaling | Adam | $10^{-4}$ | - | $10^{-5}$ |
| min-norm | Adam | $10^{-4}$ | - | $10^{-5}$ |
| DR-OMMD | Adam | $10^{-4}$ | - | $10^{-5}$ |
| OJGD | Adam | $10^{-4}$ | $10^{-4}$ | $10^{-5}$ |

The results of all algorithms on the NYU-v2 dataset are reported in Table 3 and show that OJGD obtains better or comparable results than other algorithms. Among all the algorithms, the OJGD

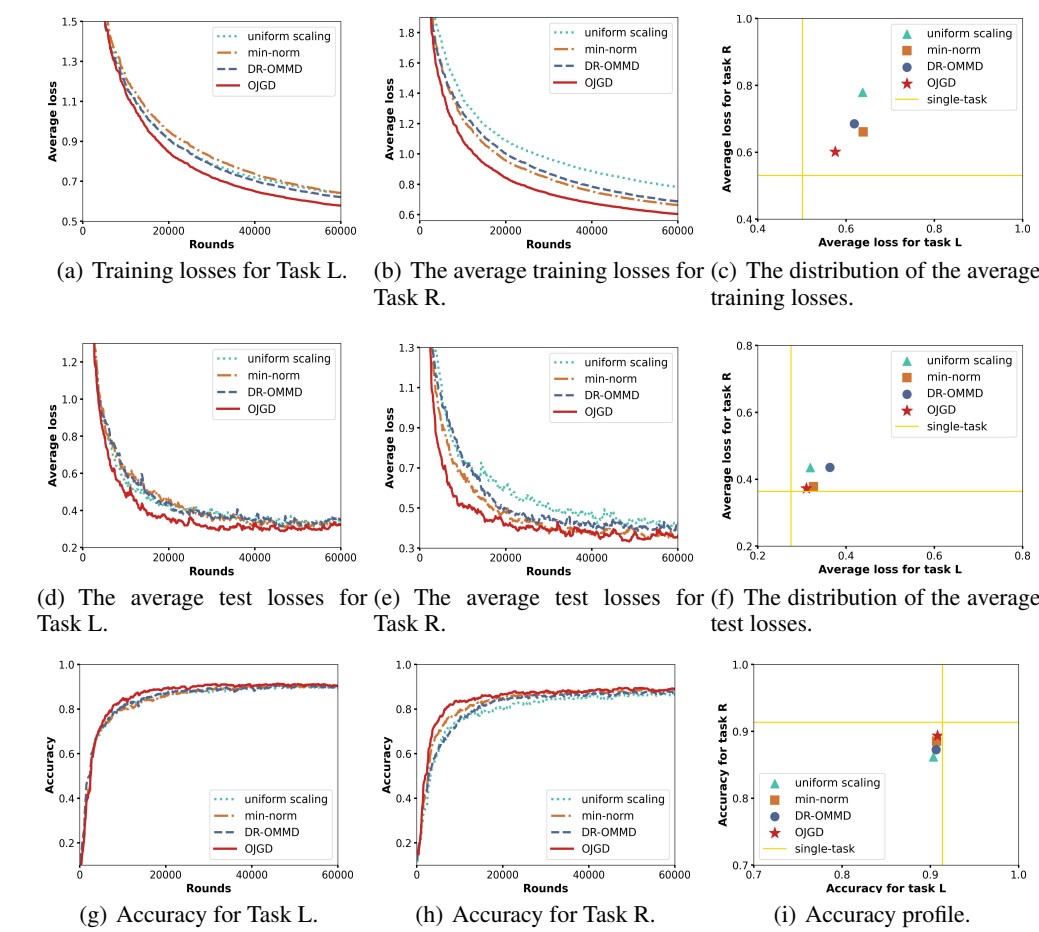

(a) Training losses for Task L.

(b) The average training losses for Task R.

(c) The distribution of the average training losses.

(d) The average test losses for Task L.

(e) The average test losses for Task R.

(f) The distribution of the average test losses.

(g) Accuracy for Task L.

(h) Accuracy for Task R.

(i) Accuracy profile.

Figure 8: More empirical results of the algorithms for two tasks on MultiMNIST. The average training losses are plotted in Figure 8(a), 8(b) and 8(c). The average test losses are shown in Figure 8(d), 8(e) and 8(f). The accuracies of the models trained by the algorithms on the test dataset are shown in Figure 8(g), 8(h) and 8(i).

algorithm achieves the best results in four single performance metrics (mIoU, Pix Acc, Abs Err and Rel Err) and the overall performance metric ($\Delta M\%$).

Table 3: Segmentation, depth, and surface normal estimation results on NYU-v2 dataset. The mean and standard deviation are reported over 3 independent runs, where ↑ indicates higher better and ↓ indicates lower better. The best average result among all the algorithms is marked in bold.

| Method | Segmentation | | Depth | | Surface Normal | | | | | $\Delta M\% \downarrow$ |
|---|---|---|---|---|---|---|---|---|---|---|
| | mIoU ↑ | Pix Acc ↑ | Abs Err ↓ | Rel Err ↓ | Angle Distance ↓ | | Within $t° $ ↑ | | | |
| | | | | | Mean | Median | 11.25 | 22.5 | 30 | |
| single-task | $52.89 \pm 0.16$ | $74.69 \pm 0.29$ | $0.4185 \pm 0.0042$ | $0.1706 \pm 0.0014$ | $21.81 \pm 0.06$ | $14.72 \pm 0.05$ | $40.03 \pm 0.11$ | $65.95 \pm 0.15$ | $75.74 \pm 0.11$ | |
| uniform scaling | $52.88 \pm 0.26$ | $74.79 \pm 0.19$ | $0.3848 \pm 0.0037$ | $0.1565 \pm 0.0036$ | $22.98 \pm 0.07$ | $16.32 \pm 0.11$ | $36.46 \pm 0.25$ | $62.46 \pm 0.20$ | $73.20 \pm 0.11$ | $1.93 \pm 0.45$ |
| min-norm | $45.87 \pm 0.15$ | $69.56 \pm 0.13$ | $0.4172 \pm 0.0034$ | $0.1707 \pm 0.0007$ | $\mathbf{21.86 \pm 0.05}$ | $\mathbf{15.07 \pm 0.15}$ | $\mathbf{39.18 \pm 0.32}$ | $\mathbf{65.24 \pm 0.33}$ | $\mathbf{75.39 \pm 0.17}$ | $2.92 \pm 0.42$ |
| DR-OMMD | $51.57 \pm 0.33$ | $73.85 \pm 0.33$ | $0.3932 \pm 0.0056$ | $0.1623 \pm 0.0040$ | $22.47 \pm 0.13$ | $15.76 \pm 0.22$ | $37.55 \pm 0.47$ | $63.79 \pm 0.51$ | $74.62 \pm 0.27$ | $1.53 \pm 0.17$ |
| **OJGD** | $\mathbf{53.11 \pm 0.28}$ | $\mathbf{75.04 \pm 0.26}$ | $\mathbf{0.3812 \pm 0.0032}$ | $\mathbf{0.1552 \pm 0.0022}$ | $22.98 \pm 0.20$ | $16.39 \pm 0.15$ | $37.12 \pm 0.41$ | $63.01 \pm 0.29$ | $73.47 \pm 0.46$ | $\mathbf{1.41 \pm 0.14}$ |

Moreover, to measure the performance efficiency per unit time of each algorithm, we report the average running time $\Delta t$ of each algorithm for a single epoch and the Performance-Time (P-T) ratio ($\frac{1/\Delta M\%}{\Delta t} \times 100$) in Table 4. Combining the results in Tables 3 and 4, the OJGD algorithm can achieve the best overall performance among all the algorithms in the same time.

Table 4: The results for the performance of the algorithm and running time on NYU-v2 dataset.The mean and standard deviation are reported over 3 independent runs, where ↑ indicates higher better and ↓ indicates lower better. The best average result among all the algorithms is marked in bold.

| Method | $\Delta M\% \downarrow$ | $\Delta t(s) \downarrow$ | P-T $\uparrow$ |
|---|---|---|---|
| uniform scaling | $1.93 \pm 0.45$ | $\mathbf{73.18 \pm 0.61}$ | $0.7351 \pm 0.1851$ |
| min-norm | $2.92 \pm 0.42$ | $278.93 \pm 3.47$ | $0.1247 \pm 0.0176$ |
| DR-OMMD | $1.53 \pm 0.17$ | $281.45 \pm 0.60$ | $0.2339 \pm 0.0263$ |
| OJGD | $\mathbf{1.41 \pm 0.14}$ | $78.26 \pm 1.18$ | $\mathbf{0.9130 \pm 0.0884}$ |

## F    THE USE OF LARGE LANGUAGE MODELS (LLMS)

In this work, we used a large language model (LLM) as an assistive tool for language polishing. We clarify that while the LLM was involved in this way, all content ultimately attributed to the authors has been reviewed, verified, and edited by the authors. We take full responsibility for all parts of the manuscript. This usage of the LLM is disclosed here and in the submission form according to ICLR 2026 policies.

