# OpenReview forum: "Online Multi-objective Convex Optimization: A Unified Framework and Joint Gradient Descent"
_ICLR.cc/2026/Conference — ICLR 2026 Conference Withdrawn Submission_

### Official Review · Reviewer_mgg6 · 2025-10-25

**Soundness:** 2
**Presentation:** 3
**Contribution:** 2
**Rating:** 4
**Confidence:** 4

**Summary:**

This paper investigates online multi-objective convex optimization (OMCO), where a multi-objective optimization problem has no unique optimal solution, but a set of efficient solutions that do not dominate each other. The authors first show that the regret of OMCO is equal to that in classical OCO when the number of objectives decreases to one. Furthermore, they propose an Online Joint Gradient Descent algorithm, which achieves a sublinear multi-objective regret by the upper bound of regret. Finally, they also conduct experiments to validate the effectiveness of their proposed algorithm.

**Strengths:**

Overall, this paper is clearly written. The authors propose a novel multi-objective metric that improves upon previous work (Jiang et al., 2023). Moreover, the multi-objective regret can be reduced to the classical online setting regret. Finally, the experimental evaluation is thorough and convincing.

**Weaknesses:**

This paper is primarily theoretical. However, the theoretical support is not entirely convincing. Regarding Theorem 1, I believe its theoretical contribution may not fully justify the designation of a theorem, and it might be more appropriate to present it as a proposition. In addition, my main concern lies in the significance of the proposed multi-objective regret and the corresponding OJGD algorithm. More specific issues are listed in the **Questions** section.

**Questions:**

**Q1:** In Theorem 2, the authors present an equivalent form of the multi-objective regret. However, minimizing this metric is essentially equivalent to optimizing a weighted regret with arbitrarily chosen weights, due to $\min_{\lambda}$. Therefore, the proposed multi-objective regret may lack intrinsic significance.

**Q2:** In the algorithmic development, the authors only consider minimizing the upper bound of the multi-objective regret in Eq. (9) rather than the original metric. It is evident that Eq. (8) represents a worst-case multi-objective target. Hence, it is unclear whether the proposed OJGD algorithm, which aims to optimize this worst-case formulation, provides substantial value.

---

> ### Author Response · Authors · 2025-11-14
> **Response to Reviewer mgg6**
>
> 1. It should be noted that the equivalent form presented in Theorem 2 is rigorously derived from the original definition of multi-objective regret (Definition 4) using convex optimization theory. This equivalent form primarily serves to provide manageable mathematical tools for algorithm design and theoretical analysis. The core significance of multi-objective regret remains embedded in its original definition. It geometrically quantifies the difference between the cumulative loss vector of an algorithm and the weakly non-dominated point set along the direction $e$, thereby authentically reflecting the degree to which the algorithm's output approximates the Pareto front.
> 2. At the algorithm design level, the comparator $x$ in the original regret metric represents a hindsight optimal solution that cannot be directly optimized during online learning. Therefore, we treat the known cumulative loss term in the original expression as fixed, thereby transforming the optimization objective into the form described by problem (9). This transformation is based on a reasonable approximation of the original metric, aiming to construct an alternative objective that can be effectively optimized in an online setting. Our proposed OJGD algorithm is designed precisely around this actionable optimization objective, and its practical effectiveness has been systematically validated through experiments on multiple real-world datasets in the paper.

---

### Official Review · Reviewer_6Pcp · 2025-10-30

**Soundness:** 2
**Presentation:** 2
**Contribution:** 1
**Rating:** 2
**Confidence:** 3

**Summary:**

This paper studies online multi-objective convex optimization (OMCO). A regret definition is proposed, derived from Translative scalarization. The authors show that this regret recovers the classical regret in online convex optimization (OCO) when the number of objectives is one. They further propose an algorithm that optimizes both action and the objective weights, and show that this algorithm achieves sub-linear regret under some assumptions.

**Strengths:**

The paper is clearly structured. The core proofs are clear and appear technically correct on a first pass. The dynamic regret is provided in the appendix.

**Weaknesses:**

Limited novelty compared to [Jiang et al, 2023]. I found the theoretical contribution of this paper to be weak. The proposed regret is essentially the same as [Jiang 2023], see Proposition 1, under convex conditions where we can intechange min and max. Besides, the algorithm seems to be a specialization of mirror-descent–style methods with Euclidean geometry (happy to be corrected if the authors think otherwise). The paper should make the precise relationship explicit and clarify what is genuinely new.

Unconvincing discussion of “non-negative regret”. This paper claims in line 128-131 that [Jiang 2023] restrict the regret to be non-negative and positions this work as removing that restriction. In my opinion, this argument is weak since regret is almost non-negative. If negative values are possible under the authors' definition, the paper should provide an example demonstrating why this is meaningful.

Theorem 1 shows that multi-objective regret is equal to the regret in OCO framework when p=1. This is very simple, expected, and does not advance understanding of the multi-objective case.

The experiments need more details. For example, in Fig 1 (a), how is the average regret computed at each round? How to compute the optimal \lambda^* at each round?

**Questions:**

How to compute the average regret in Fig 1(a)?

What is the technical novelty compared to [Jiang 2023]?

In Assumption 2, is it the loss to be convex instead of its gradient?

---

> ### Author Response · Authors · 2025-11-14
> **Response to Reviewer 6Pcp.**
>
> 1. The average regret shown in Figure 1 is straightforward to compute because, in the convex experiment, we can explicitly construct and solve an optimization instance in the form of problem (5). This practicality is a key advantage of our approach compared to the work by Jiang et al. (“Multi-Objective Online Learning”), where the regret is defined as
> $R_T=inf_{\epsilon\geq 0}\epsilon$, s.t. $\forall$ $x\in X*$, $\exists i\in[p]$, $\sum_{t=1}^{T}$$F_t^i(x_t)$-$\epsilon<\sum_{t=1}^T$$F_t^i(x)$,
>     where $X^*$ is the set of non-dominated solutions. Their regret is defined under a strict inequality constraint and a comparator limited to non-dominated solutions rather than the weaker non-dominated set, which renders it incomputable in practice even in convex settings.
>
> 2. First, as shown in the response to question 1, our regret is computable.
>
>     Second, in practice, negative regret can occur. As demonstrated by our experimental results in Appendix B, if the algorithm's decisions across multiple rounds happen to be optimal for one of the soon-to-be-revealed objectives, negative regret may arise.
>
>     Third, it theoretically unifies the online convex optimization framework for arbitrary numbers of objectives.
> 3. Thank you for your careful review. We will revise it.

---

### Official Review · Reviewer_XQ86 · 2025-10-31

**Soundness:** 3
**Presentation:** 3
**Contribution:** 3
**Rating:** 4
**Confidence:** 3

**Summary:**

This paper proposes a unified framework for Online Multi-objective Convex Optimization (OMCO), generalizing the standard Online Convex Optimization (OCO) setting to multiple objectives. The authors introduce a new definition of multi-objective regret based on translative scalarization, show that it reduces to classical regret when the number of objectives is one, and derive an equivalent dual form via convex duality. They further propose an algorithm named Online Joint Gradient Descent (OJGD) that updates both the primal decision and the objective weights jointly in an online manner. The paper provides sublinear regret bounds under standard assumptions and some empirical validation on convex regression tasks and multi-task learning benchmarks.

**Strengths:**

1. The problem itself (OMCO) is important and relevant to the ICLR community, especially with the clear connections to multi-task learning.

2. The unification of OCO and OMCO within a single regret framework is interesting and conceptually sound.

3.  The proposed OJGD algorithm is simple and computationally efficient (avoiding the QP of min-norm methods).

**Weaknesses:**

1. The paper spends a lot of time building up the new regret from Definition 4, based on translative scalarization. However, Theorem 2 immediately shows that this is equivalent to min-max problem.  This equivalent form in Eq. (6) looks very much like a standard minimax regret, i.e., finding the set of weights $\lambda$ that defines the best scalarized regret against a learner. The idea of finding the best post-hoc scalarization is not new. The unification part (Theorem 1) also feels like an expected outcome. So, the contribution in Definition 4 feels more like a (slightly complex) re-formulation of a known concept rather than a fundamentally new performance metric.

2. The algorithm is designed to solve the problem in Eq. (9), which is a classic online minimax (or saddle-point) problem. The update rules in Eq. (10) and (11) are a standard application of Online Gradient Descent-Ascent applied to the instantaneous loss $\lambda^T {F}_t(x)$. The addition of the $\alpha_t \Delta(\lambda_t)$ term is a minor modification (a regularization) to pull the weights back towards the initial $\lambda_1$. This is a well-known algorithmic template.

**Questions:**

1. Could you please clarify the novelty of the regret definition compared to the standard concept of a minimax regret (i.e., finding the optimal $\lambda^*$ on the simplex that minimizes the weighted-sum regret)? The "unification" in Theorem 1 seems to follow directly from this, so the core contribution isn't entirely clear to me.

2. How does the proposed OJGD algorithm (Eq. 10/11) fundamentally differ from a standard Online Gradient Descent-Ascent (OGDA) applied to the instantaneous game $\mathcal{L}_t(x_t, \lambda_t) = \lambda_t^T F_t(x_t)$? It looks very similar to existing primal-dual methods for online saddle-point problems.

---

> ### Author Response · Authors · 2025-11-14
> **Response to Reviewer XQ86**
>
> 1. Compared to the existing multi-objective regret proposed by Jiang et al. (“Multi-Objective Online Learning”), our multi-objective regret offers the following three advantages:
>
>    ​	1). Our regret is computable. The regret proposed by Jiang et al. (“Multi-Objective Online Learning”) is defined as
>
> $R_T=inf_{\epsilon\geq 0}\epsilon$, s.t. $\forall$ $x\in X^*$, $\exists i\in[p]$, $\sum_{t=1}^{T}$$F_t^i(x_t)-\epsilon<\sum_{t=1}^T$$F_t^i(x)$,
>
>  where $X^*$ is the set of non-dominated solutions. Their regret is defined under a strict inequality constraint and a comparator limited to non-dominated solutions rather than the weaker non-dominated set, which renders it incomputable in practice even in convex settings. However, in the convex experiment, we can explicitly construct and solve an optimization instance in the form of problem (5).
>
>    ​	   2). Considering the negative regret. As demonstrated by our experimental results in Appendix B, if the algorithm's decisions across multiple rounds happen to be optimal for one of the soon-to-be-revealed objectives, negative regret may arise. However, the regret proposed by Jiang et al. is restricted to a non-negative domain, which can't capture such negative regret phenomena.
>
>    ​	   3) Unified OCO framework for any number of objectives. It theoretically unifies the online convex optimization framework for any numbers of objectives. This establishes a more complete theoretical foundation for future algorithm design.
>
> 2. In fact,  under the OMCO framework and foundational Assumptions 1 and 2 in online settings, a standard primal‑dual algorithm requires very low dual stepsizes (weights stepsizes) to achieve sublinear regret in a static setting. However, this constraint results in performance similar to the Uniform baseline, offering little practical advantage.

---

### Official Review · Reviewer_Ggst · 2025-11-01

**Soundness:** 2
**Presentation:** 2
**Contribution:** 1
**Rating:** 2
**Confidence:** 4

**Summary:**

The paper studies Online multi-objective convex optimization by introducing multi-objective regret. The paper proposes a new multi-objective regret in OMCO based on translative scalarization that unifies the single and multi-objective frameworks. After an initial characterization of the problem, the paper proceeds to give algorithms based on the primal-dual framework. The paper is concluded by a set of experimental evaluations.

**Strengths:**

The paper tackles an important extension of online convex optimization to the multi-objective setting, proposing a unified regret definition and algorithmic framework.

**Weaknesses:**

I already reviewed this paper at NeurIPS 2025. I had included numerous points that concerned me, as well as several minor mistakes and typos. Unfortunately, none of these issues have been addressed in the updated version.

The central conceptual objection I had remains unresolved:
The paper still does not address the trivial algorithmic baseline: maintaining a separate regret minimizer for each objective and a meta-level regret minimizer that chooses which objectives to follow.

The lack of this discussion raises doubts about the necessity and novelty of the proposed method.

Also, the first half of the proof of Theorem 3 is just the standard calculations for OGD.

**Questions:**

see weknesses

---

> ### Author Response · Authors · 2025-11-14
> **Response to Review Ggst**
>
> 1. On the Meta-Expert Approach
>
>    We appreciate your suggestion of maintaining one regret-minimizer per objective, and an additional meta-regret-minimizer to select among them. However, in our specific **Online Multi-Objective Convex Optimization (OMCO)** setting, this approach faces significant practical limitations:
>
>    1)**Computational and memory overhead:** With $p$ objectives, such a scheme requires maintaining $p$ separate optimizers plus one meta-controller, and each optimizer must handle model parameter updates. When the decision variable $x_t$ corresponds to large-scale network weights and $p$ is large, this yields storage and runtime costs that scale linearly in $p$. In large-scale multi-task learning settings, this overhead becomes prohibitive.
>
>    2)**Applicability to our algorithmic setting:** Given the overhead above, this approach is not a practical candidate for our framework; it does not meet our requirements for scalability and efficiency in adversarial, time-varying multi-objective contexts. This is precisely why we did *not* adopt this scheme and why we did not include it as a baseline in our experiments.
>
> In contrast, our proposed algorithm uses a **single optimizer** (for $x$) combined with a regularized update of the dual weight vector $\lambda$. This design both (i) provides a rigorous sublinear regret bound for multi-objective regret in the adversarial OMCO setting, and (ii) achieves the best performance–runtime trade-off in practice (see Table 1 in the paper), outperforming existing OMCO methods.
>
> 2. On Theorem 3 and the OGD Computation
>
>    You correctly observe that the first part of Theorem 3’s proof uses standard online gradient descent (OGD) calculations. This is expected and deliberate: because our primal update is based on OGD (to align with widely used methods in multi-task learning), its regret bound follows known analysis templates.
>
>    As we emphasized in the paper, the contribution of the algorithm lies in the design of the update scheme for the dual variable , i.e., the weights $\lambda$. Compared to the methods based on minimizing the (regularized) norm of the joint gradient, our proposed algorithm is easier to implement and updates the weights in linear time.
>
> Finally, please be mindful of your professionalism, as I seriously question your expertise. The two issues you raised directly contradict your own line of reasoning. The first claim that our algorithm analysis is overly complex fails to consider simpler alternatives (in fact, the alternative you propose has clear practical disadvantages). The second assertion is that the OGD section's proof is standard ( in fact, we have consistently emphasized the advantages of our weight update mechanism).

---

### Note · Authors · 2025-11-18

**Comment:**

Dear PCs, ACs, and Reviewers,

We would like to express our gratitude to the esteemed reviewers for providing valuable reviews, which help enhance the quality of our manuscript. After careful consideration, we have decided to withdraw the submission to further refine our work.

Thank you for your attention to this matter.

Sincerely, Authors of OMCO-OJGD.

**Withdrawal Confirmation:**

I have read and agree with the venue's withdrawal policy on behalf of myself and my co-authors.